

# Reduction Evaluation and Management of Agricultural Non-Point Source Pollutant Loading in the Huntai River Watershed in Northeast China

YiCheng Fu*, Wenqi Peng, Chengli Wang, Jinyong Zhao, Chunling Zhang

*State Key Laboratory of Simulation and Regulation of River Basin Water Cycle, China Institute of Water Resources and Hydropower Research*

* Corresponding author, E-mail: swfyc@126.com

**Abstract:**

With the raise of environmental protection awareness, applying models to control NPS (non-point source) pollution has become a key approach for environmental protection and pollution prevention and control in China. In this study, we implanted the semi-conceptual model SWAT (Soil and Water Assessment Tool) using information on rainfall runoff, land use, soil and slope. The model was used to quantify the spatial loading of NPS nutrient total nitrogen (TN) and total phosphorus (TP) to the Huntai River Watershed (HTRW) under two scenarios: without and with projected buffer zones of approximately 1 km within both banks of the Huntai, Taizi and Daliao river trunk streams and 5 km around the reservoirs. Current land-use types within the buffer zone were varied to indicate the natural ecology and environment. The Nash-Sutcliffe efficiency coefficient ($E_{NS}$) and $R^2$ for flow and predicted nutrient concentrations (TN and TP) in a typical hydrological station were both greater than 0.6, and the relative deviation ($|Dv|$) was less than 20%. Under the status quo scenario (SQS), the simulated soil erosion in the HTRW per year was 811 kg/ha, and the output loadings of TN and TP were 19 and 7 kg/ha, respectively. The maximum loadings for TN and TP were 365 and 260 kg/ha, respectively.



Under environmental protection scenarios (EPS), the TN and TP pollutant loadings per unit
area were reduced by 26% and 14% annually, respectively. Loading analysis showed that land-
use type is a key factor controlling NPS pollution. The NPS pollutant loading decreased under
the simulated EPS, indicating that environmental protection measures may reduce the NPS
pollutant loading in HTRW. The 22% pollutant reduction under the EPS. We finally quantified
the ratio of the land area lost to agricultural production compared with that lost to ecosystem
services. We calculated the agricultural yield elasticity and concluded that the corresponding
crop yield would be reduced by 2% when the land area for ecosystem services in the basin
increased by 1% under the EPS.
**Keywords**:
Agricultural Non-Point Source pollutant loading; Environmental Protection Scenarios; Agro-
ecosystem services; Huntai River Watershed

# 36   1. Introduction

The non-point source (NPS) pollution strongly influences soil restoration, people living

environments and water quality safety. Many articles have indicated that underlying surface
conditions and rainfall features will affect the spatial distribution characteristics of NPS
pollution nutrient loading (Robinson et al.,2005). Pollutant loadings from different land-use
types vary significantly (Niraula et al.,2013). The NPS pollutant concentration in water depends
on the discharge loading and pollutant treatment rate. Presently, a lot of academics prefer
loadings over concentrations to express their study (Yang et al., 2007; Ouyang et al.,2010;
Outram et al., 2016). Land-use types and underlying surface status will impact the nutrient
resources and spatial distribution characteristics (Ahearn et al., 2005; Ouyang et al., 2013). The
spatial-temporal characteristics of NPS pollutants can be studied based on panel data statistical



analysis and multi-model simulation. The SWAT model can be implemented for NPS pollutant
loading and provide the optimization programme for comprehensive ecological protection of
watershed (Shen et al.,2011). A large number of literatures have demonstrated that combining
different land use scales, land coordinated development patterns and geomorphologic landscape
characteristics can decrease NPS pollution loadings (Sadeghi et al.,2009).

Distributed physically-based and semi-conceptual models can effectively calculate and

evaluate NPS pollution loading spatial layouts. In the late 20$^{th}$ century, American scientists at
United States Department of Agriculture-Agricultural Research Service (USDA-ARS)
developed the SWAT model (Arnold et al.,1998), which has been widely used to simulate
runoff, estimate NPS pollution loading and implement Best Management Practices (BMPs).
SWAT is comprehensively used in evaluating the influence of NPS pollution loading on
different regional natural landscape characteristics and land-use types, including vegetation
coverage, underlying surface, agricultural generation modules and hydrometeorology data. The
changes of agricultural NPS contaminations based on the diversification of land development
types have been analyzed and researched by SWAT models (Ficklin et al.,2009; Shen et al.,
2013). The SWAT model's main body contains 701 mathematical equations and 1013
intermediate variables, which have been widely used to determine and evaluate NPS pollutant
loading spatial distribution characteristics to quantitative the impacts of land use on NPS
pollutants and soil-water loss sensitive evaluation on a watershed scale (Gosain et al.,2005;
Ouyang et al., 2009; Logsdon et al.,2013).

HTRW is the basic product manufacturing base in China and a primary tributary of the Liaohe

River Basin, which has been heavily polluted in recent years. The main NPS pollution in the
Liaohe river is agricultural NPS pollution, and most NPS pollution occurs in the HTRW within



Liaoning province (Liaoning Province DEP, 2011). Therefore, the HTRW faces enormous
pressure from water pollution risk. The annual mean growth of Gross Domestic Product (GDP)
in the Liaohe River Basin was more than 13%, and the urbanization rate was almost 75%. The
policy of 'Revitalization of Old Industrial Bases in Northeast China' has caused great spatial
pattern changes to the land-use (Liu et al.,2014). This accelerating urbanization changes the
current land use in a way that results in more NPS pollution to local surface waters.
The SWAT model was applied to quantify the TN and TP output loading in HTRW under
different land uses, assess the NPS pollutant loading reduction, and analyse the spatial
distribution characteristics under the condition of land cover change. Using SWAT, nutrient
losses were simulated and evaluated under two scenarios: the status quo scenario (SQS, without
buffer zones) and the 'environmental protection' scenario (EPS, with buffer zones). We studied
NPS pollution problems in HTRW according the following steps and illustrated in the Fig. 1:
(1) define the underlying surface (land-use) status for HTRW; (2) implement a SWAT model
to simulate the NPS pollution loading (TP and TN) of the HTRW under two scenarios; (3)
compare the NPS pollution loading under the two scenarios and assessed the effect of reducing
pollutant loading under EPS; and (4) analysed the correlation between arable land area
decrease/agro-ecosystem services increase and crop yield reduction using a simple static model.



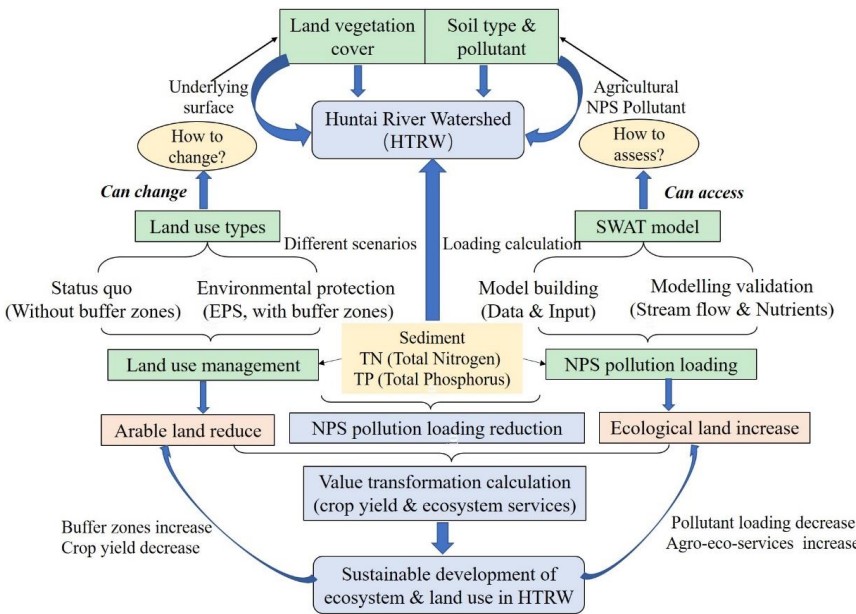


**Fig. 1. Reduction assessment and value transformation system for agricultural NPS loading**. The solid thick arrows indicate the degree of influence or the process output. The thin arrows indicate the process input.

## 2. Materials and methods

### 2.1 Huntai River Watershed

The HTRW (40°27'~42°19'N, 121°57'~125°20'E) is situated in the Liaoning province (Northeast China), and the river basin area is 2.73×10⁴ km², which comprises approximately 1/5 of the Liaoning province (Fig. 2). The HTRW is a tributary of the Liaohe River (one of China's larger water systems) and consists of the Hunhe, Taizi, and Daliao Rivers. The Taizi River, Hunhe River, and Daliao River watersheds are the HTRW's sub-catchments. The HTRW varies topographically with low mountains in the eastern portion and alluvial plains in the other areas. The northeast region has a high elevation. Loamy soils are mainly distributed in the alluvial plain, and the average slope in the lower HTRW is



approximately 7%. The HTRW area consists of the cities of Shenyang, Fushun, Liaoyang, Anshan,
Benxi, and Yingkou, most of Panjin city. Stream flow and nutrients were measured on five monitoring
stations: Beikouqian, Dongling Bridge and Xingjiawopeng in the Hunhe River and Xialinzi and
Tangmazhai in the Taizi River. HTRW has a temperate continental climate with an average annual
temperature of 7°C and precipitation of 748 mm.
The HTRW is in a conventional agricultural farming (farming-dominated products), with
much of the farmland dominated by crops. The total farmland area is 10,763 km$^2$ (39% of the
total area), including 4,086 km$^2$ of paddy fields (dominated by rice) and 6,677 km$^2$ of dry
farmland (including corn, soybean, vegetables and other crops). The upper Hunhe and Taizi
river areas are mountains (69%), plainlands (25%) and low hills (6%). The HTRW's economic
output value is dominated by agricultural cultivation. The farmland is mainly distributed in the
alluvial plain area and valleys in riverine belts. Considering land use, pollutant sources and
rainfall, the HTRW faces a high risk of agricultural pollution. Heavy fertilizer use and soil
erosion in the upper HTRW has led to its heavy water pollution. For example, the Dahuofang
reservoir (located in the middle reaches of Hunhe River) and the water resources-environment
conservation area in its upper sections are facing serious threats, and the agricultural NPS
pollution is becoming increasingly severe with no effective controls. Fertilization in the HTRW
is predominantly nitrogen, followed by potassium and phosphorous. Heavy use of chemical
fertilizers includes mainly DAP (diammonium phosphate), urea and a small amount of N-P-K
(nitrogen-phosphorus-potassium mixed fertilizer). Acetochlor and Atrazine are mainly used on
dryland, and Butachlor is mostly used in paddy soil. Based on 2006-2012 statistical information,
the fertilizer and pesticide quantities (such as Methamidophos and Plifenate) used in the
watershed fluctuated annually. The upper portions of the Huntai and Taizi Rivers are dominated





by mountains, and the crops are cultivated and harvested by hand. We obtained these data and
information, which would normally be inaccessible, through onsite investigations, inquiry visits,
case studies, example analyses. At present, farmland weeds and pests are mainly controlled by
pesticides and herbicides. The upstream is rich in forest resources, and the downstream has a
large amount of farmland. Special landscape layout makes the HTRW a potential area for
agricultural NPS pollution.

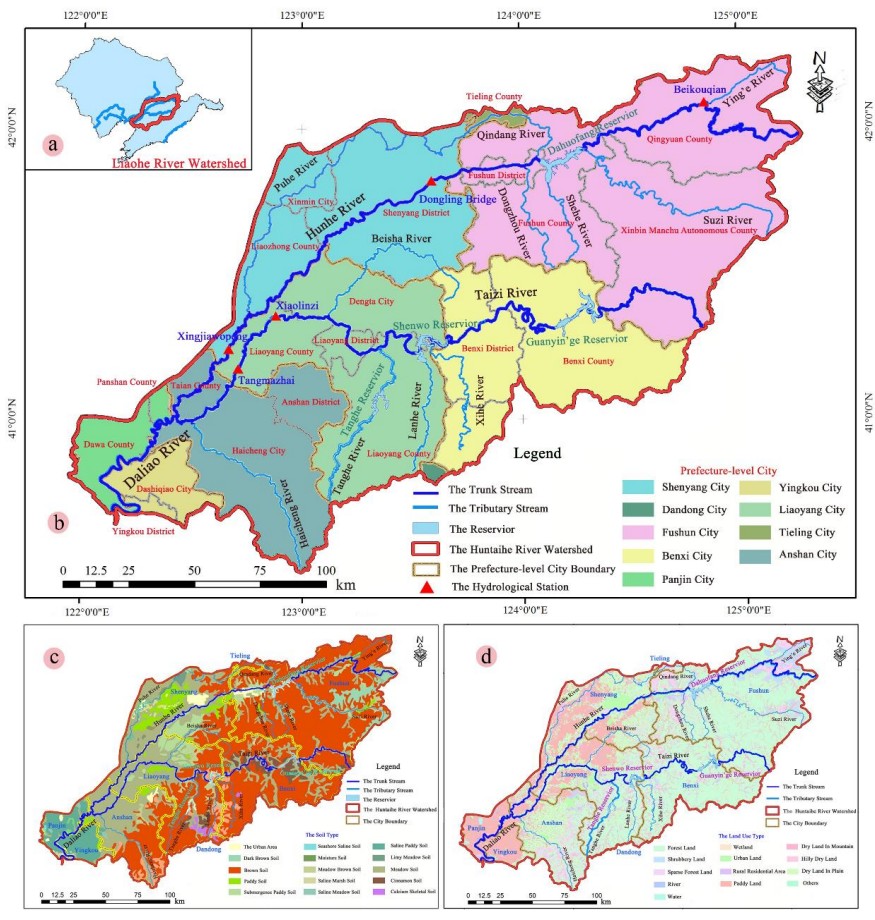

a. The location of the HTRW.    b. The geographical zoning of HTRW.

c. The land use type of HTRW.    d. The soil type of HTRW.




**Fig. 2.** Basic HTRW information. The figure was supplied by www.geodata.cn, which is a
national science and technology basic conditions platform and an earth system science data
sharing platform. The figure information is public. The Liaoning province Water Resources
Administrative Bureau granted permission for the basic information on the HTRW.
## 2.2 Setting the scene
To determine the correlation between land use types and agricultural NPS pollutant loading, the
numerical analysis and comprehensive comparison method was used for different land use types
under ecological development and urbanization. In this study, two scenarios were developed: The
status quo scenario (SQS) and the environmental protection scenarios (EPS).
The SQS was draw up based on the current environmental protection mode and
socioeconomic developmental structure. The land-use was based on the existing development
pattern and environmental protection policies. BMPs information and environment-friendly
land-use patterns (amount of pesticide and fertilizer used, cultivated land area, and crop species)
were gained from Liaoning Province statistical yearbooks-2013 and field surveys of land
consolidation.
The EPS was defined as considering the regional developmental prospects, eco-friendly
environment restoration strategy of the HTRW. Buffer zones were defined as One kilometer
within both banks of the Hunhe, Taizi and Daliao rivers and 5 km around the reservoirs. In the
buffer zones, traditional land-use patterns were changed to restore the natural landscape (forest
and grassland) and ecological environment. This scenario is not only expected to preserve the
fundamental agricultural position in the watershed but also to improve the watershed's
ecosystem service value and biodiversity by reducing the amount of fertilizers and pesticides
used for agricultural productivity. These scenarios can provide a scientific basis for further
understanding characteristics of the nitrogen and phosphorus loading characteristics and



cultivated field plantation potential adjustment in HTRW.
To simulate the hydrological characteristics by SWAT, we first divided the HTRW into a set
of sub-basins based on DEM data. We then divided sub-basins into Hydrological Response
Units (HRUs). Hunhe River, Taizi River, and Daliao River sub-catchments were delineated into
DEM and river system and further divided by 29 small calculation modules based on 184 HRUs.
We used the monitored data to calibrate and validate the stream flow and pollutant
concentration changes in the HTRW. The land development patterns in the two scenarios were
then input to the SWAT model to simulate the TN and TP pollutant loading. Finally, the spatial
dynamics and ecological service value assessment in NPS pollution loading was analysed based
on land-use. We also analyzed the negative correlation between the agro-ecological value and
the farmland area.
The wastewater pollutant source is along both channels of the Taizi, Hunhe, and Daliao River
trunk streams. The risk of NPS pollution is mainly related to the patterns of farmland use and
agricultural planting. The secondary region for water pollution is mainly along the HTRW
tributaries. Therefore, we paid special attention to the comparative analysis of pollutants
generated by the cultivated field adjacent to the water channels.
## 2.3 Methods
### 2.3.1. SWAT principle
SWAT is a semi-physical and distributed hydrological model developed to quantitatively
predict the responsivity of water quality and quantity to land-use and environmental protection
methods on a watershed scale (Gassman et al.,2007). The primary data imported to run the
model includes soil type, vegetation status/land landscape, DEM (Digital Elevation
Model)/topography, and BMPs. The watershed SWAT model's computing units are the sub-





watershed scale and HRUs. Hydrological response unit demarcation is based on land use,
vegetation coverage, soil classification, and different underlying surfaces status.
SWAT HRUs are automatically divided by geomorphological features, land development
intensity change, DEM, and soil conditions. To calculate the HRUs, we selected 0% land use,
slope/elevation, and soil classification/attributes as the initial value on the HTRW scale.
Therefore, 184 HRUs were delineated to determine NPS pollutant loading. HRUs are the
minimum units for predicting pollutant output loading, which is automatically generated by
superimposing land-use and soil types within the sub-river basin. Due to the HTRW's large
area and widely changing terrain slope, the HTRW was divided into three levels with slopes of
10 and 30 nodes. The area threshold percentages for land use, soil and slope were 5%, 8%, and
15%, respectively. To evaluate pollutant loss and spatial characteristics, the soil nutrient loss
curve, water-salt balance equation, and stage-discharge curve were applied. Meteorological
data (such as rainfall and wind speed) were gained from automatic weather stations and
hydrological station network in 12 cities within the HTRW. BMP data, such as crop irrigation
time and water, crop harvesting period, fertilizer recovery efficiency, fertilizer dosage, and
spatial layout of the overall land use planning were obtained from environmental and
agricultural management departments or collected by current questionnaire survey.
SWAT is mostly used to evaluate N and P pollutants production, diffusion and movement,
and transformation. These pollutions whole process control occur simultaneously with the soil
erosion, hydrological cycle and reasonable utilization of land resource. SWAT considers 5
forms of N and 6 forms of P. The N and P cycles contain mineralization, decomposition,
solidification/stabilization, and conversion. The NPS pollutant loading function is the basis for
evaluating N and P distribution, transportation and transformation (Zhang, 2005). Organic N





and P loss was calculated by SWAT by the comprehensive evaluation model of the NPS
pollutant loading, variations of nutrient elements and salt contents of soil, soil environment,
crop growth, and crop yield. The total amount of nitrate lost in the soil was calculated by the
multiplication of water volume and nitrate concentration in the water. Water volume consisted
of groundwater runoff, surface runoff, and interflow/subsurface flow/groundwater recession
flow. The soluble P removed in the runoff was estimated using the P concentration in the soil
partitioning coefficient, surface soil layer, and runoff volume. The concentration of soluble P
in the water was calculated by topsoil P stocks, runoff variation and influencing factors, soluble
P ratio, and soil particle-size characteristics.
Surface runoff from daily rainfall data and land use in HRU/sub-basin were calculated and
evaluated using the SCS-CN method. Using the SCS-CN curve, vertical distribution
characteristics and temporal stability of soil water, runoff module number of the ground water,
Soil saturated water content movement and hydraulic conductivity were determined, as well as
the related parameters for daily rainfall. The total discharge temporal variations of runoff from
sub-basin/HRUs is the dynamic equilibrium of groundwater runoff flow, surface runoff flow,
and interflow/subsurface flow. The main routes for water cycle simulation in the SWAT follow
either the network-node mode or the natural-artificial dualistic water cycle mode in river basins
under changing conditions. We used the dualistic mode SWAT flow varies with the dynamic
changes in infiltration, evaporation, transport, and nutrient cycling (Arnold et al.,1998). Direct
runoff is surface runoff resulting from rainfall, which includes surface and return flows.
Baseflow is part of the groundwater recharge to river runoff. Most of the base flow and direct
runoff separation methods are performed by mathematical methods. We used Digital-Filter-
Equation to divide the base flow:



$$\begin{cases} q_t = \beta.q_{t-1} + \alpha(1+\beta)(Q_t - Q_{t-1}) \\ b_t = Q_t - q_t \end{cases}$$
(1)

Here, $q_t$ is the surface runoff at time $t$; $Q_t$ is the total runoff at time $t$; $b_t$ is the base flow at time $t$;
and $\alpha$ $\beta$ are filter parameters.
Digital filtering is an objective and effective method of base-stream separation. We assigned
$\alpha$ =0.5 and $\beta$ =0.925 in the HTRW (Arnold & Allen,1999). The SWAT HRUs used the soil and
water loss factors, hydrodynamic process of soil erosion, and the universal soil loss equation
(MUSLE) to analyse erosion and sediment form, space distribution characteristic and
influencing factor (Williams, 1975). Sediment was routed through channels using Bagnold's
sediment transport equation (Bagnold, 1977). We used a 2009 version of SWAT to calculate the
parameters.
**2.3.2. Model data input**
DEM, underlying surface status, geomorphology, soil properties, land vegetation,
hydrological and meteorological data (rainfall, evaporation, temperature) were imported into
the SWAT (Niraula et al.,2013). Fig. 3 shows the basic data used in the SWAT model. We used
30×30 grid data (elevation) as the basis for DEM operation. We downloaded the DEM data for
the HTRW location from the SRTM (Shuttle Radar Topography Mission) data pack. These free
data can be obtained from the website, http://srtm.csi.cgiar.org/SELECTION/inputCoord.asp.
The DEM was used to extract the study area and analysis the stream network structure in
relation to geomorphologic features. The stream network in the HTRW was extracted using
1:250,000 digital water system data (www.geodata.com) as an auxiliary model to construct the
HTRW stream network model. We delineated land-use types into 27 categories. The main type
of HTRW land use and land cover change is forest (including orchard, 49%), dry land (24%),
rice paddy (15%), urban land (vacant land, 8%), unused land (uncultivated land, 3%) and



grassland (1%). Soil types were classified into 26 types. The primary soil types are brown soil
(54%), meadow soil (30%) and paddy soil (11%). The underlying substrate database was
constructed based on the soil type database using the soil properties and land development data
as underlying substrate parameters (topography characteristics, surface vegetation and soil
types and distribution characteristics). The soil parameters were got from national earth system
science data sharing infrastructure database (http://www.geodata.cn/aboutus.html). The
watershed meteorological data used in the current research include rainfall data for 1990-2009
collected by 76 rainfall stations/hydrometric network and hydrometeorological data for 1990-
2009 obtained by 12 city meteorological stations. We used meteorological monitoring data to
simulate rainfall and evaporation. The missing meteorological information can be estimated
using the Long Ashton Research Station Weather Generator (LARSWG-5). At least 3 sets of
water monthly monitoring data for ammonia ($NH_3$, $NH_4$), nitrite ($NO_2$), nitrate ($NO_3$), TP, and
TN, were available for 2006–2009. We obtained information on plant species, cropping systems,
sowing time, fertilization time, distribution pattern of soil productivity, and regional economic-
social development from investigations and the statistics department in HTRW. The SWAT
uses the LH-OAT (Latin Hypercube One-factor-At-a-Time) sensitivity analysis method and the
SCE-UA (Shuffled Complex Evolution Algorithm) automatic calibration analysis method to
determine the value of sensitive parameters.
Data information (scale, type, description) for SWAT in the HTRW are shown in Fig. 3. We
imported the related soil and meteorological data for SWAT from the China Meteorological
Administration and Environmental-Ecological Science Data Center for West China. The China
Hydrology, Water Resources and Water Quality Monitoring Department of the HTRW
provided the automatic and regular monitoring hydrological data sequence. The Liaoning




Province Water Resources Administrative Bureau granted permission for the modelling the
pollutant generation response to different land utilization scenarios in the HTRW.

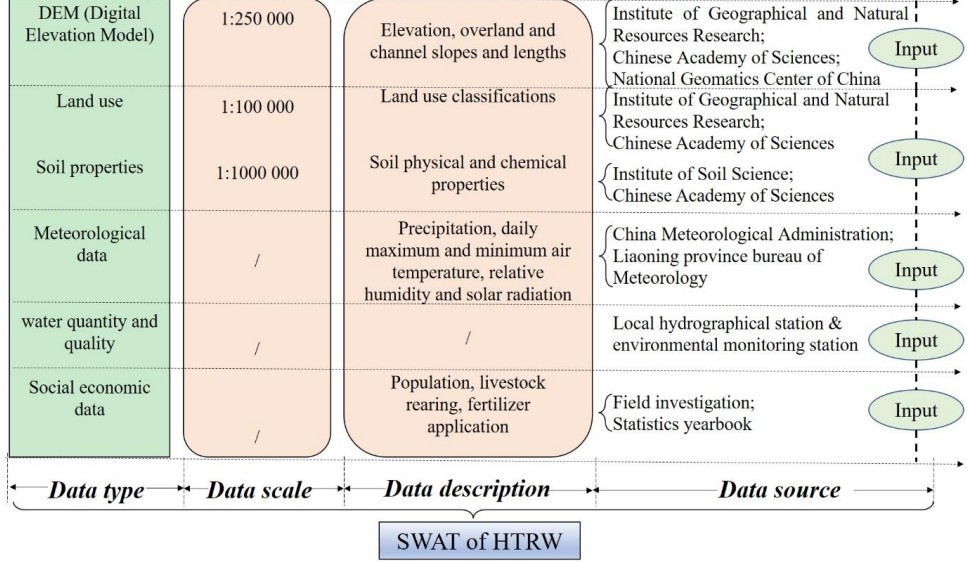


**Fig. 3.** HTRW data. The basic data imported into the SWAT model included spatial data and
attribute data. Spatial data includes DEM, land use and land cover data, soil spatial distribution
data, digital river network data, and the spatial location of meteorological stations and
hydrological stations. Attribute data mainly includes land use attribute database, soil type
attribute database, LARSWG-5 and hydro-meteorological data.

### 2.3.3. Calibration and validation

The monthly scale data were used to simulate SWAT. We used the open code SWAT-CUP
module to calibrate the parameters of SWAT in HTRW (Abbaspour et al.,2007). A sequential
uncertainty fitting algorithm had a higher calculation accuracy and efficiency, which was
extensively used in the SWAT-CUP module (Wang et al.,2014; Yang et al.,2008). We manually
input the optimal parameters into the SWAT model for hydrology series simulation. The $E_{NS}$
(Nash-Sutcliffe efficiency coefficient), $Dv$ (relative deviation), and $R^2$ (certainty coefficient)





were used to assess the runoff flow change of the HTRW hydrological station.

The runoff was calibrated, followed by N, P and other nutrients. The runoff was calibrated

and tested using monitoring data from the Xingjiawopeng and Tangmazai hydrological stations
(Fig. 2). The simulated values of N and P were calibrated using on-site monitoring data from
Dongling bridge, Beikouqian, Xiaolinzi, Xingjiawopeng, and Tangmazhai hydrological
stations. We automatically calibrated 10 sensitivity parameters, then we applied the SWAT
manual calibration helper to make small and targeted adjustments to the calibration results to
improve the simulation accuracy based on auto-calibration results. Various water quality and
hydrologic parameters (test data) were adjusted under their change interval to fit with the
monitored/observed data (Fig. 4). GWQMN, SURLAG, and ESCO were three key parameters
in the calibration and water flow validation (Shen et al., 2010). The other sensitive parameters
selected for calibration and validation in the HTRW are shown in Fig. 4. In the HTRW, the
Liaoning Province government began monthly monitoring of pollutants in 2006. The TN and
TP loading, and runoff data, used for calibration and validation were from 1992 to 2009 and
from 2006 to 2008, respectively.

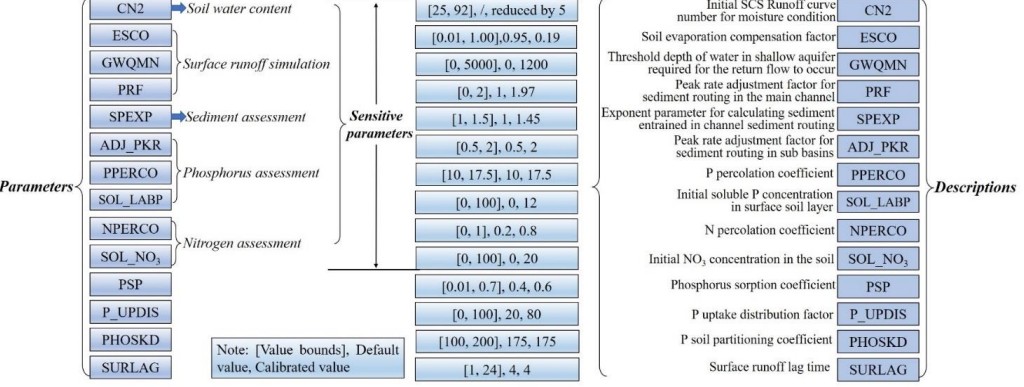

**Fig. 4.** Calibration of SWAT model parameters in the HTRW. Based on the spatial analysis of





sensitive parameters, we analyzed the influencing factors of the parameter values combining
with the underlying surface runoff factors of the basin.

In the present study, the simulated effects were evaluated by analyzing and comparing the

runoff hydrograph, $Dv$, $E_{NS}$ and $R^2$. The $Dv$ was used to simulate the entire water quantity
deviation; $E_{NS}$ and $R^2$ were used to simulate the simulation effects (Nash, 1970). The $Dv$, $E_{NS}$
and $R^2$ were calculated as

$$D_v = [(M - W)/W] \times 100\%$$                    (2)

Here, $D_v$ was the relative deviation; $W$ was the observed mean value; and $M$ was the predicted mean
value.

$$E_{NS} = 1 - [\sum_{i=1}^{n} (W_i - M_i)^2 / \sum_{i=1}^{n} (W_i - \overline{W})^2]$$                    (3)

Here, $E_{NS}$ was the Nash-Sutcliffe efficiency coefficient; $W_i$ was the observed data at the $i^{th}$
period; $M_i$ was the simulated data at the $i^{th}$ period; and $\overline{W}$ was the observed mean value.

$$R^2 = \{[\sum_{i=1}^{n} (W_i - \overline{W})(M_i - \overline{M})]/[\sqrt{\sum_{i=1}^{n}(W_i - \overline{W})^2} \sqrt{\sum_{i=1}^{n}(M_i - \overline{M})^2}]\}^2$$                    (4)

Here, $R^2$ was the certainty coefficient; $W_i$ was the observed value at time $i$; $M_i$ was the simulated
value at time $i$; $\overline{W}$ was the observed mean value, and $\overline{M}$ was the predicted mean value.

The first four years (1990-1994) were regarded as the stage for SWAT to minimize the

uncertainty of initial meteorology and underlying surface values. Sensitivity analysis of the
parameters is an effective mean to help assessing impact of uncertainty in the input and
parameters on the output uncertainty. The sensitivity evaluation indicators differed between
SWAT and SWAT-CUP. Student's t-test is used by SWAT-CUP and is part-sensitive. To
improve the model calibration accuracy and verification results, we used SWAT-CUP and the
SUFI-2 algorithm to analyse the parameters' sensitivity. To determine the sensitivity of various





parameters, one parameter was auto-adjusted at a time based on the accuracy and change
interval in Fig. 4. To calibrate the stream flow, we subsequently calibrated runoff and nutrients
(TP and TN) with the same geographical and hydrological data. During calibration, we used $R^2$
and the correlation coefficient of the residual sequence (*SCR*) to eliminate the uncertainties
caused by the differences in water quality sampling and testing methods.

## 3. Results and discussion

### 3.1 Modelling validation

**Stream flow.** Because of HTRW lacks basic runoff data, the present study focused on
calibrating and testing the runoff model. First, we dealt with the meteorological data and
retained the 1990-2001 data series, then supplied the meteorological data simulation value from
1990 to 2001 by SWAT. Second, we input the runoff data for 1995-2001 into the SWAT-CUP
model to calibrate the runoff parameters and entered these parameters into the SWAT database,
then extended the meteorological data series to 1990-2009 and simulated runoff again. Finally,
we compared the runoff simulation values with monitoring values from 2002 to 2009. During
annual calibration, the runoff curve data were calibrated and the available water content in the
soil and the soil evaporation compensation coefficient were modified. Finally, the monthly
runoff curve was modified. CN2 is a comprehensive parameter that reflects the watershed
characteristics before rainfall and is mainly affected by the hydrology and soil types, land use,
pre-soil moisture and tillage management measures. CN2 directly affects the surface runoff,
and the larger the CN2 value, the larger the runoff yield. The same land-use type yields greater
permeability with a smaller CN2 value or lower vegetation coverage and rainfall interception
ability with a greater CN2 value. Different HRUs have different CN2 values. The moist area





(climate division) has the highest CN2 ranging from 60~96, while other regions vary greatly.
Within the same soil types, the CN2 value was the highest for cultivated land, followed by
grassland. Woodland was the lowest. For the simulation, 1990-1994 was the model preparation
period, 1995-2001 was the model calibration period, and 2002-2009 was the model validation
period.
For the calibration step, $E_{NS}$ and $R^2$ for Xingjiawopeng hydrological station and Tangmazhai
hydrological station were both greater than 0.6, and the $|Dv|$ values were less than 20% during
the model preparation period, suggesting that the SWAT model parameters were reliable after
calibration. The monitoring value fit better with the simulation value obtained from
hydrographic curve. Most top values observed were highly similar. For the model calibration
period, the matching curves for the simulated and measured monthly runoff values at
Xingjiawopeng and Tangmazhai hydrological stations are shown in Figs. 5(a) and (b). The
runoffs at these two hydrological stations were well matched. However, the accuracy of the
simulated runoff for the second halves of the years 2002, 2005 and 2006 was poor, likely due
to the data series length and specific stations selected. For the simulation and evaluation
standards for the hydrological model, the simulation effects at the monthly scale were much
better.





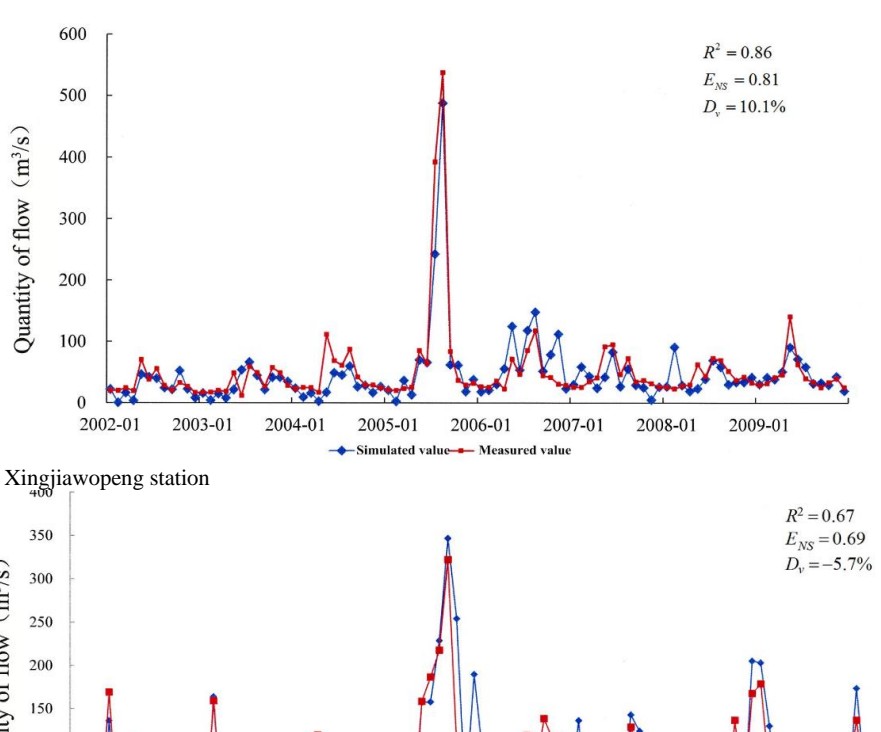

a. Xingjiawopeng station

b. Tangmazhai station

**Fig. 5.** Stream flow validation of a typical monitoring station.

**Nutrients.** The nutrient concentrations in the water were simulated by SWAT. By verifying

the accuracy of the initial concentrations, the nitrate and soluble P loading can be simulated by

adjusting the nitrogen permeability coefficient (NPERCO) and the phosphorous permeability

coefficient (Lam et al., 2011). Beikouqian, Xingjiawopeng, Xiaolinzi and Tangmazhai

hydrological stations had continuous monthly water quality monitoring data from 2006 to 2007

(model calibration period). Only the monthly data on TN and TP in Beikouqian were validated

from 2008 to 2009. The Xingjiawopeng, Xiaolinzi and Tangmazhai Hydrological stations had

only the TN data during the study period; therefore, Beikouqian was selected for the validation
curves, and the TN $E_{NS}$ and $R^2$ were 0.64 and 0.78, and the TP $E_{NS}$ and $R^2$ were 0.60 and 0.75,
respectively (Figs. 6 a and b). The $E_{NS}$ and $R^2$ for the Xingjiawopeng, Xiaolinzi and Tangmazhai
hydrological stations were 0.62 and 0.73, 0.61 and 0.72, and 0.62 and 0.77, respectively. The
values of all $R^2$ were higher than 0.7, confirming that the SWAT could be used for water quality
simulation in HTRW.

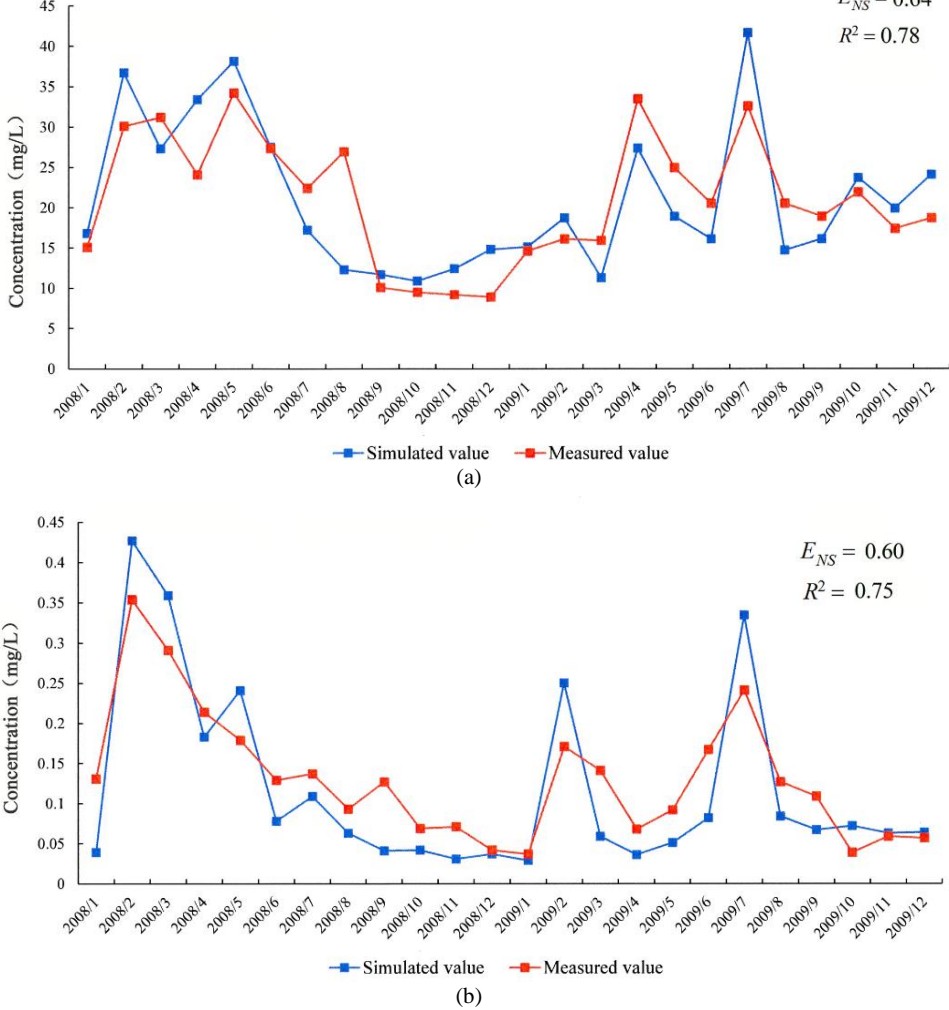





**Fig. 6.** Nutrient validation at Beikouqian station. Figure (a) and (b) shows the fitting result of TN and TP, respectively.

## 3.2 NPS pollutant loading under the status quo scenario

The NPS pollutant generation output was calculated using the pollutant loading approach based on the attributes of the regional calculation results and land-use scenarios in HTRW. The generated N and P for different calculation units were calculated based on the spatial changes in soil types, crops and residuals, as well as the differences in the coefficients of N and P losses under different land uses. The paddy fields, rural residential areas, urban development, and vegetation type may be important indicators for variability in NPS pollution, and nutrition pollution was influenced by the integrated effects of different land uses (Cai et al., 2015; Lee et al.,2010). The annual generated TN and TP were 18,707 t and 53,322 t, respectively (Table 1). Brown soil is widely distributed in the HTRW. We supplied the N and P loss characteristics under different land-use types and fertilization, as shown in Table 2 (Hao, 2012). The brown soil thickness was 30-50 cm in HTRW. The organic content, TN and TP decreased significantly with the soil depth increment. Nutrients were mainly found in soils of 0-30 cm deep, where TN and TP reserves reached more than 50% of the total soil reserves. Large-scale use of fertilizers (DAP, N:46.4% and N-P-K, N:15%; $P_2O_5$:15%; $K_2O$:15%) and livestock and poultry excrement (N:0.5-0.6%; P:0.45-0.6%; K:0.35-0.5%) were the important sources of agricultural NPS pollution. In HTRW, the number of pastures and cattle was small, and cattle excrement was collected and processed by the farmer. The excessive or unreasonable application of fertilizers and the fertilizer utilization rate were not high (the utilization rate of nitrogen is 30% to 60% and of phosphorus is 2% to 25%), resulting in substantial fertilizer loss. The nutrient content (mainly from agricultural production activities) of the soil (20cm below





the surface) in the HTRW was 1.21 t/ha. Information on initial soil nutrient content and
fertilizers was used for model parameterization.
**Table 1.** Pollutant generation in the HTRW under the status quo scenario

| Watershed | Area (km²) | Run off (E+08 m³) | Pollutant (t) | | | Pollutant loading (kg/ha) | | |
|---|---|---|---|---|---|---|---|---|
| | | | Sediment | TP | TN | Sediment | TP | TN |
| Hunhe River | 11,565 | 24.04 | 220,004 | 8,993 | 24,264 | 190 | 8 | 21 |
| Taizi River | 13,903 | 33.31 | 1,699,996 | 6,399 | 19,010 | 1,223 | 5 | 14 |
| Daliao River | 1,913 | 1.60 | 300,002 | 3,315 | 10,048 | 1,568 | 17 | 53 |
| Total/Average | 27,381 | 58.95 | 2,220,002 | 18,707 | 53,322 | 811 | 7 | 19 |

Source: China Hydrology; national earth system data sharing infrastructure; field investigation of Liaoning
province; chemical fertilizer/land area/soil erosion statistics yearbook of Liaoning province; Liaoning
province Bureau of Meteorology.
**Table 2.** Loss characteristics of N and P under different land uses and fertilization

| Land use | Soil thickness (cm) | Organic matter content (g/kg) | Unit weight of soil (g/cm³) | Soil particle composition (mm) | | | TN (g/kg) | TP (g/kg) |
|---|---|---|---|---|---|---|---|---|
| | | | | Clay $\varnothing \leq 0.002$ | Loam $0.002 < \varnothing \leq 0.005$ | Sand $0.005 < \varnothing \leq 2$ | | |
| Cultivated field | 0-5 | 24.58 | 1.42 | 21.05 | 57.35 | 21.6 | 0.96 | 0.47 |
| | 5-30 | 18.45 | 1.48 | 24.71 | 24.71 | 18.84 | 0.88 | 0.38 |
| | 0-5 | 27.6 | 1.18 | 15.97 | 15.97 | 14.58 | 1.25 | 0.58 |
| Grassland | 5-30 | 21.75 | 1.25 | 20.36 | 20.36 | 21.5 | 1.02 | 0.42 |

## 3.2.1. Sediment

Sediment loading is the data basis for calculating TN and TP loading and is affected by the
type of land development and vegetation coverage (generally dominated by forest and
farmland). Based on the SWAT model simulation, the annual sediment outputs generated in the
Hunhe, Taizi and Daliao River watersheds were $22 \times 10^4$ t, $170 \times 10^4$ t and $30 \times 10^4$ t, respectively.
The annual soil erosion loading in HTRW was 0.811 t/ha, and its spatial distribution is shown
in Fig. 7(a). The soil erosion value varied widely in different regions, with the change interval
from 0 to 2 t/ha. Soil erosion in the Daliao River watershed was severe (up to 2 t/ha in some



regions), followed by the Taizi River watershed (1 t/ha in most regions) and the Hunhe River
watershed (less than 0.2 t/ha in most regions). Yingkou and Dashiqiao have even topography,
and incoming silt from the upper areas is accumulated therein. The soil erosion modulus is
therefore very high, which contributes greatly to the silt input to the HTRW. The soil erosion
was affected by natural and human factors. The natural factors mainly included topography,
underlying surface conditions and soil type. The human factors mainly consisted of vegetation
coverage, precipitation, land use, crop cultivation and cultivated land farming methods.
Moreover, mountainous areas have great soil erosion (Hong et al.,2012). The Daliao River had
a large cultivated land area; therefore, soil erosion was more likely. Soil types are also key
influencing factors in causing soil erosion; therefore, brown and paddy soils are prone to
accumulating sediment (Hong et al.,2012).

## 3.2.2. TP

From the SWAT simulation results, the annual TP output generated in the Hunhe, Taizi and
Daliao River watersheds was 8993 t, 6399 t and 3315 t, respectively, and the HTRW output
loading was 7 kg/ha. The TP loading had the same spatial distribution pattern as the sediment
loading, ranging from 0 to 260 kg/ha. Fig. 7(b) shows the TP spatial variation loading of the
HTRW. A large amount of P could be deposited in the downstream plain. The changes in the
TP loading were affected by topography, precipitation, land use, and silt losses. The TP output
loading on the Daliao River watershed slope was higher than that of the Hunhe River watershed,
while the Taizi River watershed was the lowest. Many fertilizers and pesticides have been
applied to the farmland, and organophosphate pesticides accounted for 40% of the total
pesticides (Wang, 2012). The paddy fields, brown soil and dry lands were mainly distributed in
the Hunhe River downstream. Therefore, the P loading in these plains areas was higher (Li et



al., 2010). Correspondingly, the cities and counties with large proportions of farmland have
higher TP output loading, such as Dashiqiao, Panshan and Dawa city in the Daliao River
watershed and the city of Haicheng and Taian county in the Hunhe River watershed. Regions
with large proportions of developed land have lower TP output loading, including the city centre
of Fushun, Shenyang in the Hunhe River watershed, the municipal districts of Liaoyang city
and Benxi city at the Taizi River watershed. Based on land use, tributaries with higher
proportions of farmland have the highest TP output loading, while tributaries with substantial
vegetation cover as forested land have relatively lower TP output loading. TP output loading is
closely related to soil characteristics and attributes.
**3.2.3. TN**
Simulation and calculation results showed that the TN generation output in the Hunhe, Taizi
and Daliao River watersheds was 24264 t, 19010 t and 10048 t. The annual TN output loading
in the watershed was 19 kg/ha. Fig. 7(c) shows the spatial variation of TN loading in the HTRW.
The TN loading interval varied from 0.001 to 365 kg/ha. The TN loading had the same spatial
characteristics as TP loading. The TN output loading in the Daliao River watershed was greater
than that in the Hunhe River watershed, while the Taizi River watershed was the lowest. Large
amounts of fertilizer were applied in the study area. Nitrate and organic N accounted for a
substantial portion of the fertilizer used in HTRW. Therefore, the TN output loading in the
watershed was very high. Regions with much farmland, such as the middle and lower portions
of the Hunhe River, the lower portions of the Taizi River and the tributaries in the upper portions
of the Daliao River, have high TN output loading. The organic N contents in the forested land
were very low. Thus, the output loading of TN in regions with high vegetation forest cover,
such as the mountainous areas in the upper parts of the Taizi and Hunhe rivers, was very low.



The TN output loading in municipal districts with highly developed areas was the lowest, such

as in the municipal districts of Fushun city and Shenyang city in the Hunhe River watershed

and the municipal districts of Benxi city, Liaoyang city and Shenyang city in the Taizi River

watershed.

TN and TP loading in the HTRW were characterized by a regional distribution. Although

Qingyuan, Yibin and Benxi counties, located in the upper areas of the HTRW, had high water

and silt output, their pollutant loading was low. Per unit area, the maximum TN and TP loading

(maximized over space) was 365 and 260 kg/ha, respectively. The regions TN and TP with high

loading were mainly distributed in Taian, Haicheng, and Fushun city. The TP and TN loading

near the Dahuofang, Tanghe, Shenwo and Tanghe reservoirs were low, ranging from 0.006 to

10 kg/ha and from 0.08 to 19 kg/ha, respectively. Based on topography and soil distribution,

the slope is very steep in the upper stream of HTRW. The soils are predominately brown soil

and salted paddy soil, both of which are easily eroded. The topography in the lower sections is

usually flat, such as in the cities of Anshan, Haicheng, Yingkou and Panjin. The elevation is

low, and the soil is predominately meadow soil and brown soil, both of which have higher soil

erosion rates, silt loss and output loading of pollutants. The regions with heavy TN and TP

loading included Xinmin county, located in the middle and lower reaches of the HTRW, the

municipal district of Shenyang city, Liaozhong county, Dengta city, Liaoyang county, the

municipal district of Anshan city, Haicheng city and a portion of Dashiqiao city. Based on the

land development pattern in the Taizi River, dry fields and paddy fields were mainly distributed

on the plain areas of this watershed, which constitute therefore a core source of output loading.

The spatial differences in the TN and TP loading were no large differences. Based on

topography, landform, soil types and land development status in the watershed, the upper





streams of the watershed have high vegetation coverage, less farmland and low pollutant
loading, while the lower areas have more farmland, high fertilizer application rates and high
soil erosion and pollution loading (Yin et al.,2011). In summary, the spatial characteristics of
TN loading resulted from comprehensive effects of precipitation/runoff characteristics, soil
properties, soil erosion and vegetation coverage. Therefore, to effectively control TN loading
and soil erosion in the HTRW, the BMPs, fallow measures of cultivated fields, watershed
vegetation restoration and soil and water conservation in the upper stream are the most
important measures to implement.

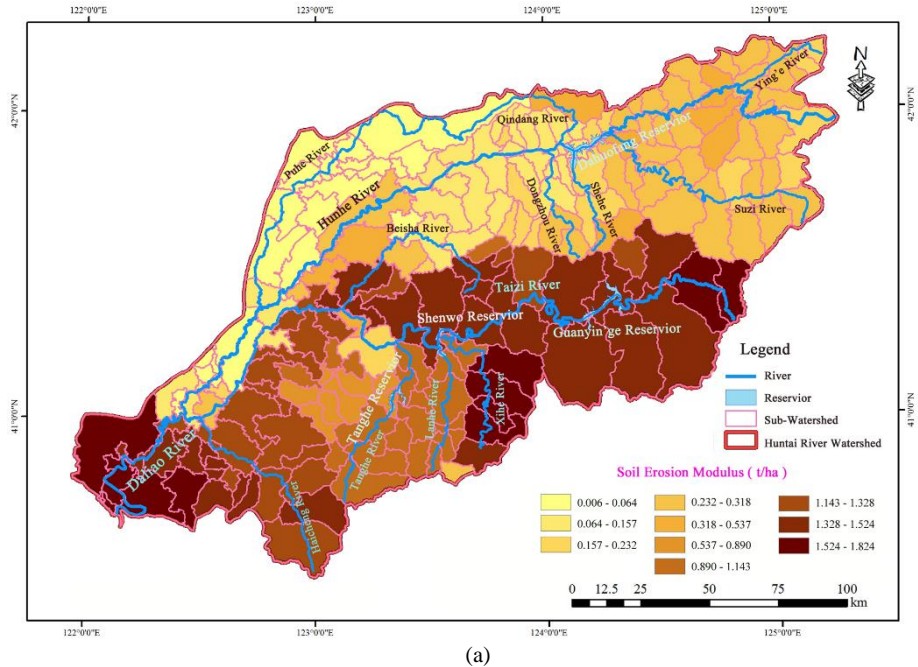

(a)







(b)

(c)

**Fig. 7.** NPS pollution loading distributions of HTRW under the status quo scenario. The Figure





(a), (b) and (c) showed the loading distributions of contaminant sediment, TN and TP, respectively.

## 3.3 NPS pollutant loading under EPS

There exists a correlation between land development mode and water environment protection and rehabilitation at the basin scale. The riparian buffer zones effectively reduced the concentration levels of $NO_3^-$ in the water, which were 47% lower than those of the soil content. Dry farmland caused higher NPS pollutant loading, followed by paddy lands, rural and urban areas, forestland, and shrub lands. Hence, under the EPS, the farmland area in the watershed was reduced. A modest area of farmland (29,500 ha, accounting for 3% of the total farmland area) was converted to forestland (including shrub land, 14,753 ha; grassland, 5,899 ha; and wetland, 8,848 ha), while NPS pollution from farmland decreased. The water quality protection objective within the watershed's critical zoning was realized. The riparian buffers can be planted in various diverse vegetations. The N removal rate of a 60-m-wide woody soil buffer zone was 16% and 38% higher than that of shrubbery and grassland, respectively. Approximately 1 kilometre within both banks of the Hunhe, Taizi and Daliao river tributaries and 5 kilometres of surrounding reservoirs buffer zones were defined, including 1946 km$^2$ of farmland, urban land, and rural residential land. This accounts for 7% of the total area in the watershed. The woodland coverage rate was reduced by 1%, and the loading of sediment, TP and TN increased by 0.01-11, 0.2-3 and 0.4-14 kg/km$^2$, respectively. The pollutant generation output under EPS was calculated by transforming the existing land-use type.

The TN and TP respective ranges of change were 0 to 365 kg/ha and 0 to 260 kg/ha. The annual losses of TN and TP were reduced by 13,839 and 1,946 t/yr., respectively. In comparison, the NPS pollutant generation output under the EPS was decreased by 22% compared with that



under the SQS, whereas the TP and TN outputs were reduced by 10% and 26%, respectively.
Under the EPS, the average loading of TN and TP was 14 and 6 kg/ha on a unit area basis,
which were 14% and 26% less than the loading under the SQS, respectively. The NPS pollutant
loading declined in the EPS. The variation of TP and TN pollutant loading between the SQS
and EPS is shown in Table 3. The amount of change indicated that riparian buffer and land
development pattern change effectively reduced NPS pollutant loading in the HTRW.
**Table 3.** Loading variation in TP and TN pollutant between EPS and the status quo scenario

| Watershed | Pollutant loading of EPS (kg/ha) | | Pollutant loading variation (kg/ha) | | Farmland variation (ha) | Forestland variation (ha) | Grassland variation (ha) | Wetland variation (ha) | Pollutant annual variation(t/yr.) | |
|---|---|---|---|---|---|---|---|---|---|---|
| | TP | TN | TP | TN | | | | | TP | TN |
| Hunhe River | 7 | 16 | -1 | -5 | -12,460 | +6,231 | +2,492 | +3,737 | -838 | -5,743 |
| Taizi River | 4 | 10 | -1 | -4 | -14,979 | +7,491 | +2,995 | +4,493 | -776 | -5,606 |
| Daliao River | 16 | 40 | -1 | -13 | -2,061 | +1,031 | +412 | +618 | -332 | -2,490 |
| Total/Average | 6 | 14 | -1 | -5 | -29,500 | +14,753 | +5,899 | +8,848 | -1,946 | -13,839 |

"—" denotes a decrease compared to that of the status quo scenario; "+" denotes an increase compared to
that of the status quo scenario.

## 3.4 Value transformation based on NPS pollutant loading

Based on the results of Simpson (2014), the correlation between the yield, input of land,
purchase investment, ecosystem services value, and land area is expressed as
$$q = f(x, S, A) \qquad (6)$$
where $q$ is the yield, $x$ is the quantity of the purchase investment, $S$ is the quantity of the
supplied ecosystem services, $A$ is the land area directly utilized for agricultural production.
Formula (6) is a conceptual formula, and there is no sole unit for each letter in formula (6).
To obtain a general expression, we give a special instance of production function.
$$q = f(x, S, A) = x^\alpha S^{1-\alpha} - \gamma (x^\alpha S^{1-\alpha})^2 / A \qquad (7)$$
Where $\alpha = 1/2$, $\gamma$ was a positive constant.
The constant profit of a massive production function is expressed as (Vincent & Binkley,

1993)




$$q = \sqrt{x \cdot S} - \gamma \cdot x \cdot S / A \qquad (8)$$

This is a simple production function with limitations. Assuming that r represents the price of
the input. When the price of the yield is normalized as 1, the profit $\eta$ is expressed as
$$\eta = \sqrt{x \cdot S} - \gamma \cdot x \cdot S / A - r \cdot x \qquad (9)$$

For x, the first-order condition for maximal profit is expressed as
$$\begin{cases} \dfrac{1}{2}\sqrt{S/x} - \gamma \cdot S / A - r = 0 \\ -\dfrac{\varphi}{2}\sqrt{x/S} + \gamma \cdot x \cdot \dfrac{\varphi \cdot \overline{A}}{A^2} = 0 \end{cases} \qquad (10)$$

and when the first-order of Eq. (10) is satisfied, the second-order condition for maximal
management was also satisfied. After the conversion, we obtained
$$x = \frac{S}{4(\gamma \cdot S / A + r)^2} \qquad A = \frac{\overline{A}}{1 + \sqrt{r / \gamma \cdot \varphi}} \qquad (11)$$

The ecosystem services of the preserved land were determined by the parameter, $\varphi$. In the
production function, ecosystem services and the purchase investment were the substitute items.
When the purchase price increases, the area of the land used for production will decrease,
suggesting that more land should be preserved. When more land is preserved for maximal profit,
the yield will decrease. When Eq. (10) was multiplied by x, we obtained
$$\frac{1}{2}\sqrt{x \cdot S} - \gamma \cdot x \cdot S / A - r \cdot x = \eta - \frac{1}{2}\sqrt{x \cdot S} = 0 \qquad (12)$$

where the total differential of Eq. (12) was calculated concerning A. If A and x were used to
realize maximal profits, the arithmetic resolution could be calculated as
$$\frac{\mathrm{d}x / x}{\mathrm{d}A / A} = \frac{A}{\overline{A} - A} \qquad (13)$$

therefore, when most of the land was used for production, the dependence on the purchase
investment increased. If more land was used in agricultural production activities, greater costs



would be paid to compensate for lost ecosystem services. If the initial consideration was the
excessive dependence on purchase investments, the margin rate of technically replacing the
purchase investment by ecosystem services increased, indicating that the purchase investment
would be reduced significantly.
If the land area used for agricultural production was changed, profits ($\partial q/\partial x=r$) were
maximized by
$$\frac{\mathrm{d}q/q}{\mathrm{d}A/A} = \frac{r \cdot x}{q} \frac{A}{\overline{A}-A} \tag{14}$$

If the agricultural production was dependent on the purchase investment in the current
watershed to a greater degree, a significant yield reduction would occur.
The total value of the water and soil conservation, material investment, and service
investment was calculated using the energy price per unit of land area in the river basin as a
reference (Fu et al., 2017). The agricultural production investment value in HTRW was
calculated indirectly by the equivalent conversion between the investment and the acquired
value. The cost investment for agricultural production in HTRW was ¥ 217.13 E+08
(USD$ 34.12 E+08) (Table 4).
**Table 4.** Energy prices of agricultural production in HTRW (based on status quo scenario).

| Classification | Details | Energy value (Sej/ha) | Price of energy (¥/ha) | Total values (¥ E+08) | Cost investment (¥ E+08) |
|---|---|---|---|---|---|
| Water and soil conservation | Soil erosion | 3.08E+14 | 594.53 | 6.40 | 6.40 |
| Material investment | Depreciation | 3.05E+15 | 1,960.34 | 21.10 | 21.10 |
| | Fuel oil | 2.90E+13 | 18.64 | 0.20 | 0.20 |
| | Material | 1.42E+16 | 9,126.83 | 98.23 | 98.23 |
| | Labour | 1.29E+16 | 8,291.28 | 89.24 | 89.24 |
| Service investment | Maintenance and management | 2.51E+14 | 161.33 | 1.74 | 1.74 |
| | Service | 3.30E+13 | 21.21 | 0.23 | 0.23 |
| *Total* | | | | 217.13 | 217.13 |

To calculate the elasticity of the farmland yield data derived from Eq. (13), the expression
was used as follows:





$$\left(\frac{\mathrm{d}q}{q}\right) / \left(\frac{\mathrm{d}A}{A}\right) = \left(\frac{217.13E+08}{3800E+08}\right) / \left(\frac{29500}{1076300}\right) = 2 \,. \tag{15}$$

Here, $\mathrm{d}q$ was the farmland production cost after deducting the paid amount, $q$ was the
agricultural production value of the farmland, $\mathrm{d}A$ was the total reduced area of the farmland
under EPS, and $A$ was the total area of the farmland. We obtained the agricultural production
value from Liaoning Province statistical yearbooks-2013.
Based on this calculation, to maximize profit, 1% of the currently cultivated or used land in
HTRW was converted into preserved land to supply more ecosystem services. Accordingly, the
crop yield in HTRW was reduced by 2%. The analysis results based on the ecological land-use
value in HTRW, the forestland, grassland, and wetlands were the main suppliers of ecosystem
services. Therefore, the decreased farmland and increased the ecological value of the non-
cultivated land were profit indicators.
## 4. Conclusions
NPS pollution often occurs on dry farmlands and in paddy, rural and urban areas. Many
studies have applied the SWAT model to study NPS in China, mainly focusing on scenario
simulations of NPS pollution and management in agricultural areas with rich hydrological and
meteorological data. Basic monitoring data on HTRW were deficient. Thus, we selected SWAT
as a feasible method for assessing NPS pollutant loading at the watershed level. We applied
specific practices based on EPS to reduce NPS pollutant loading in the Hunhe, Taizi and Daliao
River watersheds. The status quo scenario and EPS were used to calculate the NPS pollutant
generation output. NPS pollutant generation output and TN and TP loading were reduced by
22%, 26% and 10% compared with that of the SQS, respectively. The crop yield was reduced
by 2% when the land area for ecosystem services in the basin increased by 1%.





The SWAT model can calculate the potential reduction of agricultural NPS pollutants based
on different land uses. The reliability of SWAT evaluation results are decided by information
completeness and the reasonable degree of parameter initialization. To determine pollutant
reduction under different land development patterns and examine the uncertainty of
sensitivity parameters, the SWAT model in China has many potential applications.
Considering the significance of ecosystem services, much attention should be paid to the
relationship between ecosystem service increases and crop yield decreases. Only in this way
can ecosystem services and land use be harmoniously developed.

## Acknowledgements

The study was financially supported by the National Key Research and Development Program
of China (2016YFC0401408), Comprehensive Regulation Theory and Application of Basin
Water Environment Process (WE0145B532017) and Project Based Personnel Exchange
Program with China Scholarship Council & German Academic Exchange Service of 2015.

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
