# Peer review of "Reduction Evaluation and Management of Agricultural Non-Point"

_Hydrology and Earth System Sciences, 2018_

## Short Comment (SC1) · 13 Aug 2018

Generally, the manuscript addresses an important topic. The work in the manuscript is sufficient to be a publication. However, the writing needs to be improved in some sections of the manuscript. Please see specific comments below. Abstract: Please write full words of abbreviations before using them. For example NPS, SWAT in the abstract. The authors should check abbreviations throughout the manuscript. L16: "The study topics is mainly focus on", correct to "The study topic mainly focuses on". The purpose of the study is very general. I prefer specific objectives of the study. L17-18: " SWAT model was constructed based on rainfall runoff 18 and land use type":

SWAT model also uses soil types and slope information. L20: What do you mean by systematically analyzed? Can you describe what you did? L24: What you mean by "scenario settings" in your study? In the Results and Discussion of the abstract, you should mention your results for calibration and validation before discussing about the results from scenarios.

Introduction L53-54: "The concentrate…between different areas". Grammar is not right. Please rewrite. L73-L74: "The SWAT model has 701 mathematical equations…" This is really unnecessary. The model is continuously updated and equations are continuously added. L91: "contrast the different" I guess you mean compare.

Materials and Methods Section 2.1 about description of study area is too long. Please shorten it and only mention necessary information. L141-L147 " For the calculation process … farmers status quo". I think these sentences should belong to the model setup section. The description about SWAT model is too long. Since we can find these information in many previous studies and in the manual of SWAT, there is no need to describe them in details. Please shorten it and only choose the necessary information to describe. L184-185: " We used 30×30 grid data (elevation) as the basis for DEM operation". What did you do to prepare the DEM data? L193-195 " The database of the underlying substrate was constructed based on the database of soil types using the soil properties & land development data as underlying substrate parameters". I don't understand what you want to say here. What are substrate parameters here? L204-205 "All the data were validated by the standard procedures used by the SWAT". Can you specify the standard procedures? L228-229: Which period is used for calibration, and for validation? L283-288: Your description on streamflow calibration is not clear about how you did for annual calibration and how you used the annual calibration to do monthly calibration. Did you use SWAT-CUP for this calibration?

Is the SWAT setup you used for calibration called the status quo scenario described in the Scenarios setting? L271-272: 29 smaller modeling units, are they subbasins in SWAT? Or HRUs? Then after that you mentioned 184 HRUs. But with the number of

soil types (26 types) and land use types (27 types), the number of HRUs (184) seems to be a very small number.

I think the results are valuable, however, I don't feel they have been presented well to the reader.

Conclusion I feel that the conclusion is just repetition of the results and discussion. I don't think you should repeat the number of TN and TP loads under two scenarios. You should summarize what you learn from the results and discuss about them.
* * *

---

## Short Comment (SC2) · 14 Aug 2018

After careful consideration, we feel that it has merit, but is not suitable for publication as it currently stands. Therefore, my decision is "Major Revision." General Content: There are numerous spelling mistakes and grammatical errors. This paper requires English editing and proof assistance. Authors should be careful using acronyms. If an acronym appears for the first time in the text, then it should be written in the full form. Ln 28: What is "pollutant production" mean? Did authors mean by pollutant generation? Ln 30: modulus? What does that mean, not sure why there is a mean and a modulus both? Again, could be something typo or just something else. Ln 30-31:

output intensities? Why not only output? Or load? Or Concentration? Or flux? Overall this section is also difficult to read. Ln 31: intensities? Ln 95: please do not refer readers to go and read literature from database. If there is literature pertinent to this paper, cite them; otherwise please do not direct readers. They can find any articles in Google scholar easily or other sites from libraries. Ln 83-85: there is no link between these two sentences. The last sentence may need to be a starting sentence for this paragraph instead. Ln 86: "The SWAT" of the present study"? Not sure what authors are trying to relate to? Ln 123: Inconsistency is writing proper names? For example, Atrazine is written with capital A, but not other fertilizers/pesticides such as acetachlor and butachlor? Ln 127: did the authors mean complete data? Statistics of what? Model Description: Ln 132: instead of "calculate", "predict" might be a better word. Ln 142: the threshold of 0% creates large number of HRUs to capture all heterogeneities. The reason behind using 0% threshold is not well justified. Model Inputs: Ln 188: 1:250 000, there is an additional space in 250000, it should be a comma. Calibration and Validation: Ln 216-218: why only NSE? Using NSE alone, as a performance indicator is not sufficient. It will not indicate any bias in model output. I am assuming that there are more metrics used. Ln 219: what is artificial parameter modification? Ln 221-225: What is the difference between "real data" and "monitoring data"? Aren't they both real? Ln 243-245: If the authors used SWAT-Cup for automatic calibration, then what is this manual calibration? Sensitivity analysis is included in SWAT-CUP. Did authors conduct a separate sensitivity analysis outside of SWAT-CUP? If so why? There is no need. Ln 244-250: Did the authors separate baseflow from total streamflow and calibrate runoff and baseflow separately? If so, the method needs to be clearly stated on how it was done? Manual or SWAT CUP? What is the point of using LOADEST program during the calibration. I understand that LOADEST could be used to calculate pollutant load, but what is unclear it the progam used during calibration. Ln283-299 In the preceding section, the authors mentioned that CN2 has no role. But it appears that runoff curve data was used for calibration. Please clarify. Results and Discussions: Similar observations could be made to TP and TN results and discussions. What are

the common fertilizers used in the farmlands? Are there pastures and cattle feeding lots? What are the initial soil nutrients content? Did the authors use this information for model parameterization? It is unclear from the methods and discussion. This paper could be improved if given time to rewrite everything. Paper could be published after revision.

---

## Author Comment (AC1) · 14 Aug 2018

**Dear Editor,**

We are so appreciated for your letter on our manuscript "Reduction Evaluation and Management of Agricultural Non-Point Source Pollutant Loading in the Huntai River Watershed in Northeast China", Reference No: hess-2018-339. We are also extremely grateful to the editors'/reviewers' comments on our manuscript and carefully considered every comment and made cautious revision accordingly. Based on their suggestions, we have answered the questions in detail one by one. If you have any other questions about this paper, I would quite appreciate it if you could let me know them in the earliest possible time.

Most sincerely,

YiCheng Fu, Wenqi Peng, Chengli Wang, Jinyong Zhao, Chunling Zhang
First Contact: Yicheng Fu, swfyc@126.com

Corresponding author:

Name: yi-cheng FU

E-mail: swfyc@126.com

14th, Agu. 2018

**Additive list**

We have studied the valuable comments from you, the assistant editor and reviewers carefully, and tried our best to revise the manuscript. The point to point responds to the reviewer's comments are listed as following.

**Reviewer's Responses to Questions**

Generally, the manuscript addresses an important topic. The work in the manuscript is sufficient to be a publication. However, the writing needs to be improved in some sections of the manuscript. Please see specific comments below.

(1) Please write full words of abbreviations before using them. For example, NPS, SWAT in the abstract. The authors should check abbreviations throughout the manuscript.

**Answer:** Thanks for your very thoughtful suggestion.

We have made serious changes to the expression of abbreviations in the whole paper, such as NPS (Non-point source), SWAT (Soil and Water Assessment Tool), TN (Total Nitrogen), TP (Total Phosphorus), HTRW (Huntai River Watershed), environmental protection scenario (EPS), DEM (Digital Elevation Model), and BMPs (Best Management Practices scenarios).

The revised contents could be found in the file of "paper revised version (clean)".

(2) L16: "The study topics is mainly focus on", correct to "The study topic mainly focuses on". The purpose of the study is very general. I prefer specific objectives of the study.

**Answer:** Thanks for your very thoughtful suggestion.

We have revised the "The study topics is mainly focus on" to "The study topic mainly focuses on".

In order to make the article clear, we have revised the ""The study topic" to "The focus point". This section is the application scope of SWAT model, which was not the specific objectives of the study. The study objectives of the paper was "The model was used to quantify the spatial loading intensities of NPS nutrient TN (Total Nitrogen) and

TP (Total Phosphorus) to HTRW (Huntai River Watershed) under two scenarios (without & with buffer zones). The NPS pollutant loading decreased under the EPS, which showed that environmental protection measure could effectively cut down NPS pollutant loading in HTRW. SWAT was used to assess the reduction of agricultural NPS pollutant."

The revised contents could be found in the file of "paper revised version (clean)" & paper revised version (with track changes).

(3) L17-18: " SWAT model was constructed based on rainfall runoff and land use type": SWAT model also uses soil types and slope information.

**Answer:** Thanks for your very thoughtful suggestion.

We have improved SWAT model information, and have added the soil types and slope information to the SWAT. The revised contents could be found as the followed,

"SWAT model was constructed based on rainfall runoff, land use type, soil types and slope information.".

(4) L20: What do you mean by systematically analyzed? Can you describe what you did?

**Answer:** Thanks for your very thoughtful suggestion.

The systematically analysis contained three parts, which were (1) scenarios setting of SWAT; (2) modelling validation of SWAT in HTRW; (3) NPS pollutant loading calculation under status quo scenario & EPS.

The revised section was as followed,

Besides, the loadings and distribution traits of NPS pollutants were also systematically analyzed based on the model (scenarios setting, modelling validation, and pollutant loading calculation under status quo scenario & EPS).

(5) L24: What you mean by "scenario settings" in your study?

**Answer:** Thanks for your very thoughtful suggestion.

The "scenario settings" is the mean of "Land use types differences".

The revised contents could be found in the file of "paper revised version (clean)" & paper revised version (with track changes).

(6) In the Results and Discussion of the abstract, you should mention your results for calibration and validation before discussing about the results from scenarios.

**Answer:** Thanks for your very thoughtful suggestion. We added the following contents,

The $E_{NS}$ (Nash-Sutcliffe efficiency coefficient) & $R^2$ (certainty coefficient) of stream & nutrients (TN & TP) in typical hydrological station were both greater than 0.6, and the $|Dv|$ (relative deviation) was less than 20%. The SWAT could be used in HTRW.

The revised contents could be found in the file of "paper revised version (clean)" & paper revised version (with track changes).

(7) Introduction, L53-54: "The concentrate…between different areas". Grammar is not right. Please rewrite.

**Answer:** Thanks for your very thoughtful suggestion.

We carefully devised the expression of the sentence. The revised contents were followed,

The NPS pollutant concentrate in water is dependent on the discharge intensity and pollutant treatment rate, therefore, which was difficult to make a fair comparison between different areas (Tucci 1998; Dingman 2002; de Oliveira et al.,2016).

(8) Materials and Methods. Section 2.1 about description of study area is too long. Please shorten it and only mention necessary information.

**Answer:** Thanks for your very thoughtful suggestion. We have shortened the length of Section 2.1. We only provided the necessary information of study area. The contents were been found as following,

The HTRW (40°27′~42°19′N, 121°57′~125°20′E) is in Liaoning province (Northeast China), and the watershed area is 2.73×104 km2, which takes about 1/5 of the area of Liaoning province (Fig 1). The HTRW is a tributary of Liaohe River Basin (The Liaohe River Basin is one of China's larger water systems) and is consist of Hunhe River, Taizi River, and Daliao River. The Hunhe River, Taizi River, and Daliao River watershed is HTRW's sub-watershed. The HTRW has varied topography, low mountain is located in eastern part, and the other parts are alluvial plain. The elevation of

northeast region is high. Loamy soils are mainly distributed in alluvial plain, and the average grade of lower HTRW is about 7%. HTRW area includes the cities of Fushun, Shenyang, Benxi, Liaoyang, Anshan, and Yingkou, most of Panjin city, some portions of Tieling city and a minor portion of Dandong city. The stream flow and nutrient were validated based on the five monitoring stations, Beikouqian, Dongling Bridge and Xingjiawopeng are located in Hunhe River, Xialinzi and Tangmazhai are in Taizi Rive. HTRW has temperate continental climate, the average annual temperature is 7°C, and precipitation is 748 mm.

(9) L141-L147 " For the calculation process … farmers status quo". I think these sentences should belong to the model setup section.

**Answer:** Thanks for your very thoughtful suggestion. We have put the " For the calculation process … farmers status quo" to the model setup section.

(10) The description about SWAT model is too long. Since we can find these information in many previous studies and in the manual of SWAT, there is no need to describe them in detail. Please shorten it and only choose the necessary information to describe.

**Answer:** Thanks for your very thoughtful suggestion. We have shortened the length of SWAT model description. We only provided the necessary information of SWAT model. We supplied some information of SWAT in the form of figure, such as Figure 1, and Figure 2.

(11) L184-185: " We used 30×30 grid data (elevation) as the basis for DEM operation". What did you do to prepare the DEM data?

**Answer:** Thanks for your very thoughtful suggestion.

We download the DEM data of HTRW from the SRTM (Shuttle Radar Topography Mission) data pack, the free data can be obtained on the website of http://srtm.csi.cgiar.org/SELECTION/inputCoord.asp. With GIS (Geographic Information System) platform, we obtained the DEM data of HTRW, as well as hydrological station & weather station distribution, by using the technology of DEM data projection transformation, splicing and cutting.

(12) L193-195 " The database of the underlying substrate was constructed based on the database of soil types using the soil properties & land development data as underlying substrate parameters". I don't understand what you want to say here. What are substrate parameters here?

**Answer:** Thanks for your very thoughtful suggestion.

The underlying substrate parameters means the data of topography characteristics, surface vegetation and soil types & distribution characteristics. These data were the basic to calculate NPS pollutant loading and distribution intensity changes.

(13) L204-205 "All the data were validated by the standard procedures used by the SWAT". Can you specify the standard procedures?

**Answer:** Thanks for your very thoughtful suggestion.

We added the related contents were as followed,

The SWAT uses the LH-OAT (Latin Hypercube One-factor-At-a-Time) sensitivity analysis method & SCE-UA (Shuffled Complex Evolution Algorithm) automatic calibration analysis method to determine the value of sensitive parameters.

The revised contents could be found in the file of "paper revised version (clean)" & paper revised version (with track changes).

(14) L228-229: Which period is used for calibration, and for validation?

**Answer:** Thanks for your very thoughtful suggestion.

We added the related contents were as followed,

The runoff, TN & TP loadings data used for calibration & validation were from 1992 to 2009, from 2006 to 2008, respectively.

In L287, to the stream flow, "For the simulation, 1990-1994 was the model preparation period, 1995-2001 was the model calibration period, and 2002-2009 was the model validation period." The contents could be found in the file of "paper revised version (clean)" (L296-L297).

In L304-306, to the nutrients, "Beikouqian, Xingjiawopeng, Xiaolinzi and Tangmazhai four hydrological stations had a continuous monthly water quality monitoring data from 2006 to 2007. Only the monthly data of TN & TP in Beikouqian

were validated from 2008 to 2009 for the insufficient of water quality monitoring data.".
Therefore, the 2006-2007 was the model calibration period, and 2008-2009 was the
model validation period.

The revised contents could be found in the file of "paper revised version (clean)"
& paper revised version (with track changes).

(15) L283-288: Your description on streamflow calibration is not clear about how you
did for annual calibration and how you used the annual calibration to do monthly
calibration. Did you use SWAT-CUP for this calibration?

**Answer:** Thanks for your very thoughtful suggestion.

We added the related contents were as followed,

(1) First, we dealt with the meteorological data and retained the 1990-2001 data
series, then supplied the meteorological data simulation value from 1990 to 2001 by
SWAT;

(2) We input into the runoff data of 1995-2001 to SWAT-CUP model to calibrate
the runoff parameters;

(3) We took the (2) parameters into the database of SWAT, then extended the series
of meteorological data to 1990-2009 and simulated runoff again.

(4) At last, we compared the runoff simulation values with monitoring value from
2002 to 2009.

The added contents could be found in the file of "paper revised version (clean)" &
paper revised version (with track changes).

(16) Is the SWAT setup you used for calibration called the status quo scenario described
in the Scenarios setting?

**Answer:** Thanks for your very thoughtful suggestion.

The scenarios setting for calibration was called the status quo scenario.

(17) L271-272: 29 smaller modeling units, are they sub-basins in SWAT? Or HRUs?
Then after that you mentioned 184 HRUs. But with the number of soil types (26 types)
and land use types (27 types), the number of HRUs (184) seems to be a very small
number.

**Answer:** Thanks for your very thoughtful suggestion.

We added the related contents as followed,

To simulate the hydrological characteristics by SWAT, firstly, we divided the HTRW into a certain number of sub-basins according to DEM data, the sub-basins have the same characteristics of soil & land use; then we divided sub-basins into HRUs.

(18) I think the results are valuable, however, I don't feel they have been presented well to the reader.

**Answer:** Thanks for your very thoughtful suggestion.

In order to increase the readability of the paper, we reduced the number of pictures, and increased the number of tables to describe the reduction of agricultural NPS pollution loading. The spatial distribution of the mean annual TP and TN loading in the HTRW were 19, and 7 kg/ha, respectively. The region with a high NPS pollution loading is located in the middle and lower the HTRW, which included the urbanization and population density highly areas of Shenyang, Liaoyang and Anshan. Under the EPS, the TN and TP per unit area were 14, and 6 kg/ha, respectively. The output of NPS pollutant production, the loading intensities of TN & TP was reduced by 21.9%, 25.9% and 10.4% compared with the status quo scenario, respectively. The NPS pollution occurring within different sub-basins and regions located in the watersheds varied greatly, and the loading intensities of different pollutant types in the given sub-basin were slightly different. Land eco-restoration measures should be implemented to control agricultural NPS pollution from croplands. Therefore, SWAT simulation results provide a reference for the prevention of agricultural NPS pollution in agricultural watersheds.

(19) Conclusion

I feel that the conclusion is just repetition of the results and discussion. I don't think you should repeat the number of TN and TP loads under two scenarios. You should summarize what you learn from the results and discuss about them.

**Answer:** Thanks for your very thoughtful suggestion.

We have deleted the number of TN and TP loads under two scenarios. And

summarized the contents that we learn from the results and discuss. We revised the contents as followed,

The NPS pollution is prone to cause in dry farmland, paddy, rural & urban areas. The SWAT model has been applied to study NPS in China by numerous research literature, they were mainly focuses on scenario simulation of NPS pollution and management in agricultural areas with rich hydrological and meteorological data. The basic monitoring data of HTRW were deficient, we selected the SWAT as the feasible method to access NPS pollutant loading in watershed level. We applied certain practices based on EPS to reduce the NPS pollutant loading in the Hunhe River, Taizi River and Daliao River watershed. The status quo scenario and EPS were used to calculate the output of NPS pollutant production. The output of NPS pollutant production, the loading intensities of TN & TP was reduced by 21.9%, 25.9% and 10.4% compared with the status quo scenario, respectively. In different regions of NPS pollutant loading in the HTRW changes greatly, and the pollutant loading intensity of different nutrients in the same region is slightly different. Land eco-restoration and land development mode adjustment measures should be practiced reducing NPS pollutant loading of cultivated land.

The revised contents could be found in the file of "paper revised version (clean)" & paper revised version (with track changes).

*We tried our best to improve the manuscript and made some changes in the manuscript. These changes will not influence the content and framework of the paper. And here we did not list the changes but marked in red in revised paper (Revision, changes marked).*

*We appreciate for Editors/Reviewers' warm work earnestly, and hope that the correction will meet with approval.*

*Once again, thank you very much for your comments and suggestions.*

---

## Author Comment (AC2) · 17 Aug 2018

Dear editors,

We would like to submit the enclosed manuscript entitled "*Reduction Evaluation and Management of Agricultural Non-Point Source Pollutant Loading in the Huntai River Watershed in Northeast China*" (hess-2018-339), which we wish to be considered for publication in "*Hydrology and Earth System Sciences* (HESS)*"*.

No conflict of interest exits in the submission of this manuscript, and manuscript is approved by all authors for publication. In this work, we evaluated the manuscript is a part of our present research achievement, and which is a good paper. I hope this paper is suitable for "HESS". The main point our thesis wishes to address is to reflect on the practical application of and the solutions provided by the SWAT models in relation to China's sparse hydrological basin information; to provide point by point model constructions; an explanation of our process; an analysis of our results, and the expansion of the utilization of the SWAT model from an advanced and disciplined perspective. SWAT was used to assess the reduction of agricultural NPS pollutant. Buffer zone of land use type could reflect the natural environment. 21.9% pollutant reduction under the EPS.

We have tried our best to revise the manuscript to hope to meet with approval. The manuscript has been thoroughly checked again and revised as suggested with the help of an English teacher. It is believed that the revised paper will be readable and could meet the standard generally for publication.

Thank you very much for your consideration and help. Looking forward to hearing from you soon.

Thank you very much for your time and consideration.

Yours sincerely,

Dr. Yicheng Fu

E-mail: swfyc@126.com

+86-10-68781880 (office); +86-10-68572778 (fax)

Organization name: China Institute of Water Resources and

Hydropower Research (IWHR)

Organization address: A-1 Fuxing Road, Haidian District, 100038 Beijing

17th, Agu. 2018

**Additive list**

We have studied the valuable comments from you, the assistant editor and reviewers carefully, and tried our best to revise the manuscript. The point to point responds to the reviewer's comments are listed as following.

**Reviewer's Responses to Questions**

After careful consideration, we feel that it has merit, but is not suitable for publication as it currently stands. Therefore, my decision is "Major Revision."

(1)General Content: There are numerous spelling mistakes and grammatical errors. This paper requires English editing and proof assistance. Authors should be careful using acronyms. If an acronym appears for the first time in the text, then it should be written in the full form.

**Answer:** Thanks for your very thoughtful suggestion.

The manuscript has been thoroughly checked again and revised as suggested with the help of an English teacher (AJE, http://www.aje.com/). It is believed that the revised paper will be readable and could meet the standard generally for publication.

The EDITORIAL CERTIFICATE is followed.

**EDITORIAL CERTIFICATE**

This document certifies that the manuscript listed below was edited for proper English language, grammar, punctuation, spelling, and overall style by one or more of the highly qualified native English speaking editors at American Journal Experts.

**Manuscript title:**

An Estimation of Agricultural Surface-source Pollutant Production in Huntai River Watershed Based on the SWAT Model

**Authors:**

Y.C. Fu, J. Zhang, C.L. Zhang

**Date Issued:**

July 21, 2016

**Certificate Verification Key:**

B8D0-B6CF-FE85-D272-7AD3

[Figure]

This certificate may be verified at www.aje.com/certificate. This document certifies that the manuscript listed above was edited for proper English language, grammar, punctuation, spelling, and overall style by one or more of the highly qualified native English speaking editors at American Journal Experts. Neither the research content nor the authors' intentions were altered in any way during the editing process. Documents receiving this certification should be English-ready for publication; however, the author has the ability to accept or reject our suggestions and changes. To verify the final AJE edited version, please visit our verification page. If you have any questions or concerns about this edited document, please contact American Journal Experts at support@aje.com.

An Estimation of Agricultural Production of Surface Non-point source Source Pollutionant Production in the Huntai River Watershed Based on the SWAT Model

Abstract: A SWAT model was constructed matching tobased on the rainfall runoff and land use types was constructed; the migration-transformation processes of agricultural surface non-point source pollutants was were simulated and calculated, and the emission export load and distribution traits of surface non-point source pollutants were systematically analyzed based on the SWAT model. the The SWAT model was calibrated and tested by using the realactual monitoring data; as well as the physical propertiesy of the underlying surfacesubstrate, hydrology, meteorology and pollutant sources in the Huntai Reiver watershed. One kilometer within both banks of the trunk streams of the Huntai river, Taizi river and Daliao rivers and 5 km peripheral surrounding the reservoirs were defined as buffer zones. Existing land use types within the buffer zones were changed to restore the natural ecologyenvironment. The throughput of pollutant production under the regional ecology environmental protection priority mode scenario was calculated based on the conventional development modescenario. In the case of conventional development mode scenario, the annual mean modulus of soil erosion in the Huntai Reiver watershed was 400 kg.hm$^{-2}$, and the output intensitiesy of total N and P was were 19 and 7 kg.hm$^{-2}$, respectively. Seen fromFor the unit area, the maximal load intensitiesy for total N and P was were 317 and 260 kg.hm$^{-2}$, respectively. The spatial difference of tTotal N

批注 [Ed1]: Abbreviations and acronyms are often defined the first time they are used within the main text and then used throughout the remainder of the manuscript. Please consider adhering to this convention.

Besides, we have made serious changes to the expression of abbreviations in the whole paper, such as NPS (Non-point source), SWAT (Soil and Water Assessment Tool), TN (Total Nitrogen), TP (Total Phosphorus), HTRW (Huntai River Watershed), environmental protection scenario (EPS), DEM (Digital Elevation Model), and BMPs (Best Management Practices scenarios).

The revised contents could be found in the file of "paper revised version (clean)".

(2)Ln 28: What is "pollutant production" mean? Did authors mean by pollutant generation?

**Answer:** Thanks for your very thoughtful suggestion.

The mean of "pollutant production" is "pollutant generation". We have revised all the expressions in the paper. We have revised 23 places.

The revised contents could be found in the file of "paper revised version (clean)" & paper revised version (with track changes).

(3)Ln 30: modulus? What does that mean, not sure why there is a mean and a modulus both? Again, could be something typo or just something else.

**Answer:** Thanks for your very thoughtful suggestion.

The expression of "soil erosion modulus" is a proper noun.

In order to improve the readability of the article, reduce ambiguity, we have revised the sentence "the annual mean modulus of soil erosion in the HTRW was 811 kg/ha" to "the soil erosion modulus in the HTRW per year was 811 kg/ha".

In line 20, The systematically analysis contained three parts, which were (1) scenarios setting of SWAT; (2) modelling validation of SWAT in HTRW; (3) NPS

pollutant loading calculation under status quo scenario & EPS.

The revised section was as followed, Besides, the loadings and distribution traits of NPS pollutants were also systematically analyzed based on the model (scenarios setting, modelling validation, and pollutant loading calculation under status quo scenario & EPS).

In line 24, The "scenario settings" is the mean of "Land use types differences".

The revised contents could be found in the file of "paper revised version (clean)" & paper revised version (with track changes).

(4)Ln 30-31: output intensities? Why not only output? Or load? Or Concentration? Or flux? Overall this section is also difficult to read.

Ln 31: intensities?

**Answer:** Thanks for your very thoughtful suggestion.

The expression of "output intensities" is the mean of "output loading".

In order to improve the readability of the article, reduce ambiguity, we have revised the all the expressions in the whole paper, such as intensities.

The Ln28, we revised "output intensities" to "output loading".

The Ln29, we deleted the word of intensities.

The Ln35, we deleted the word of intensities.

The Ln83, we deleted the word of intensities.

The Ln57, we revised "output intensities" to "output loading".

The Ln65,66,67,68, we deleted the word of intensities.

Besides, we also revised some places such as intensity to loading, or deleted these words to make the article more fluent.

(5)Ln 83-85: there is no link between these two sentences. The last sentence may need to be a starting sentence for this paragraph instead.

**Answer:** Thanks for your very thoughtful suggestion.

We have changed the order of sentences. In order to make the expressions of the paragraph clearer, we have put the last sentence at the beginning of this paragraph.

(6)Ln 86: "The SWAT" of the present study"? Not sure what authors are trying to

relate to?

**Answer:** Thanks for your very thoughtful suggestion.

We have changed the expression of the sentence. In order to make the expressions of the content clearer, we have revised the sentence as the following,

"The SWAT Model was applied to quantify the output loading of TN & TP in HTRW under different land use types, assess the NPS pollutant loading reduction, and analyze the spatial distribution characteristics on the condition of land vegetation cover change.".

(7)Ln 95: please do not refer readers to go and read literature from database. If there is literature pertinent to this paper, cite them; otherwise please do not direct readers. They can find any articles in Google scholar easily or other sites from libraries.

**Answer:** Thanks for your very thoughtful suggestion.

We have We have shortened the length of Section 2.1. We only provided the necessary information of study area. The contents could be found as following,

The HTRW (40°27′~42°19′N, 121°57′~125°20′E) is in Liaoning province (Northeast China), and the watershed area is 2.73×104 km2, which takes about 1/5 of the area of Liaoning province (Fig 1). The HTRW is a tributary of Liaohe River Basin (The Liaohe River Basin is one of China's larger water systems) and is consist of Hunhe River, Taizi River, and Daliao River. The Hunhe River, Taizi River, and Daliao River watershed is HTRW's sub-watershed. The HTRW has varied topography, low mountain is located in eastern part, and the other parts are alluvial plain. The elevation of northeast region is high. Loamy soils are mainly distributed in alluvial plain, and the average grade of lower HTRW is about 7%. HTRW area includes the cities of Fushun, Shenyang, Benxi, Liaoyang, Anshan, and Yingkou, most of Panjin city, some portions of Tieling city and a minor portion of Dandong city. The stream flow and nutrient were validated based on the five monitoring stations, Beikouqian, Dongling Bridge and Xingjiawopeng are located in Hunhe River, Xialinzi and Tangmazhai are in Taizi Rive. HTRW has temperate continental climate, the average annual temperature is 7°C, and precipitation is 748 mm.

Besides, we have put the basic information of SWAT model, and the operation & application procedure show in the form of pictures (Figure 2, Figure 3), to reduce the length of the article, and increase the readability of the article. We also cited the relevant literature to add the scientific and readability of the paper, such as Ln 137, L138, L140, and L145, et. al.

(8)Ln 123: Inconsistency is writing proper names? For example, Atrazine is written with capital A, but not other fertilizers/pesticides such as acetachlor and butachlor?

**Answer:** Thanks for your very thoughtful suggestion.

The words of "diammonium phosphate, potassium phosphate, atrazine and acetochlor" were proper names. We have revised the form of these words in the form of special nouns. Such as

Ln 125-127, The heavy use of chemical fertilizers was mainly urea, DAP (diammonium phosphate) & a small amount of N-P-K (Nitrogen-Phosphorus-Potassium mixed fertilizer).

Ln 127-130, Atrazine & Acetochlor were mainly used on dry farmland, and Butachlor was mainly used in paddy fields. Based on the statistical data for 2006-2012, the quantity of fertilizers and Pesticides (such as Methamidophos & Plifenate) applied in the watershed fluctuated annually.

(9)Ln 127: did the authors mean complete data? Statistics of what?

**Answer:** Thanks for your very thoughtful suggestion.

We revised the ambiguous expression. The revised sentence is as follows,

"and therefore statistical department statistics (http://www.ln.stats.gov.cn/tjsj/tjgb/, http://www.stats.gov.cn/) are not available.". We obtained these data & information which would normally be inaccessible through the on-site investigation, inquiry visits, case studies, example analysis, expert consultation and material research method.

(10)Model Description:

Ln 132: instead of "calculate", "predict" might be a better word.

**Answer:** Thanks for your very thoughtful suggestion.

We have revised the word "calculate" to "predict".

(11)Ln 142: the threshold of 0% creates large number of HRUs to capture all heterogeneities. The reason behind using 0% threshold is not well justified.

**Answer:** Thanks for your very thoughtful suggestion.

We have added 0% as the basis for generating HRUs. The added contents were as follows,

HRU is the minimum unit to predict pollutant output loading, which is automatically generated by the superimposition of land use & soil types within the sub-river basin. Because lots of HRUs were automatically generated based on different combinations,we selected 0% underlying surface data as the initial value to generate HUR that is consistent with the distribution characteristics of HTRW water system.

(12)Model Inputs:

(12)Ln 188: 1:250 000, there is an additional space in 250000, it should be a comma.

**Answer:** Thanks for your very thoughtful suggestion.

We have revised the space to a comma in 250000.

We also revised the other data in the whole paper.

(13)Calibration and Validation:

Ln 216-218: why only NSE? Using NSE alone, as a performance indicator is not sufficient. It will not indicate any bias in model output. I am assuming that there are more metrics used.

**Answer:** Thanks for your very thoughtful suggestion.

We have added the relative contents as follows,

We used the open SWAT-CUP software to adjust parameters, then the SUFI-2 algorithm was selected to determine the optimal values of the parameters based on iterative computations. Finally, we manually input the optimal parameters to SWAT model for hydrology series simulation. The $E_{NS}$ (Nash-Sutcliffe efficiency coefficient), $Dv$ (relative deviation), and $R^2$ (certainty coefficient) can effectively avoid the uncertainty of hydrological sequence (precipitation, water flow, and evaporation),

which was used to evaluate the run-off flow change of hydrological station in HTRW.

(14)Ln 219: what is artificial parameter modification?

**Answer:** Thanks for your very thoughtful suggestion.

We revised the ambiguous expression. The revised sentence is as follows,

The SWAT for the present study was calibrated and tested using the coupled method of manual & auto-calibration. The uncertainty analysis was carried out by using SWAT-CUP program.

(15)Ln 221-225: What is the difference between "real data" and "monitoring data"? Aren't they both real?

**Answer:** Thanks for your very thoughtful suggestion.

In the paper, the "real data" is the "monitoring data". We have revised the ambiguous expression. The revised sentence is as follows,

The runoff was calibrated and tested using monitoring data from the Xingjiawopeng, and Tangmazai hydrological station (Figure 4).

(16)Ln 243-245: If the authors used SWAT-Cup for automatic calibration, then what is this manual calibration? Sensitivity analysis is included in SWAT-CUP. Did authors conduct a separate sensitivity analysis outside of SWAT-CUP? If so why? There is no need.

**Answer:** Thanks for your very thoughtful suggestion.

In the paper, we used the coupled method of manual & auto-calibration to analyze the parameters sensitivity. The revised sentence is as follows,

Sensitivity analysis of the parameters is an effective means to reduce the uncertainty of the hydrological model and increase effectiveness of SWAT model. The sensitivity evaluation indicators are different among SWAT and SWAT-CUP. The "T-test (Student's t test)" used by SWAT-CUP is part-sensitive. We draw on manual calibration analysis to make necessary adjustments to the SWAT-CUP sensitivity analysis. To improve the accuracy of model calibration & verification results, we used SWAT-CUP and SUFI-2 algorithm to analyze the parameters sensitivity.

(17)Ln 244-250: Did the authors separate baseflow from total streamflow and calibrate runoff and baseflow separately? If so, the method needs to be clearly stated on how it was done? Manual or SWAT CUP? What is the point of using LOADEST program during the calibration. I understand that LOADEST could be used to calculate pollutant load, but what is unclear it the program used during calibration.

**Answer:** Thanks for your very thoughtful suggestion.

In the paper, we have revised the ambiguous expression. The revised sentence is as follows,

(1) Direct runoff is surface runoff resulting from rainfall, which includes surface and return flows. Baseflow is part of groundwater recharge to river runoff. It is impossible to measure or directly divide the base flow from the total runoff. Most of the base flow and direct runoff segmentation are performed by mathematical methods. We used Digital-Filter-Equation to divide the base flow (Lyne & Hollick, 1979).

$$
\begin{cases}
q_t = \beta.q_{t-1} + \alpha(1+\beta)(Q_t - Q_{t-1}) \\
b_t = Q_t - q_t
\end{cases}
\tag{1}
$$

Here, $q_t$ is the surface runoff at time $t$; $Q_t$ is the total runoff at time $t$; $b_t$ is the base flow at time $t$; $\alpha$ & $\beta$ are filter parameter.

According to the following steps. (1) The first filter made the second record points as a starting point forward calculate one by one. We calculated $q_t$ according to equation (1), and $b_t \geq 0$, $q_t \geq 0$, $Q_t \geq b_t$, $Q_t \geq q_t$. If $q_t < 0$ at time $t$, we assigned $q_t = 0$, $Q_t = b_t$; If $b_t < 0$ at time $t$, we assigned $b_t = 0$; If $b_t > Q_t$ at time $t$, we assigned $b_t = q_t$. (2) The second filter made the penultimate record points as a starting point backward calculate based on the calculation result of first filter. (3) The calculation of third filter following the positive operation. According to the above calculation rules, we divided the base flow until getting the smooth base flow process line. Digital filtering is an objective & effective method of base-stream segmentation, we assigned $\alpha = 0.5$, $\beta = 0.925$ in HTRW (Arnold & Allen,1999).

(2) We used the Auto-Calibration & Uncertainty module (SWAT-CUP) of SWAT to automatically calibrate 10 sensitive parameters, then we applied the Manual

calibration helper of SWAT to make small & targeted adjustments to the calibration results to improve the simulation accuracy based on auto-calibration results. The contents could be found in Ln 247-250 (paper revised version (clean)).

The reference as following,

Arnold, J., Allen, P.: Automated Methods for Estimating Baseflow and Ground Water Recharge from Streamflow Records. Journal of the American Water Resources Association, 35, 411-424, 1999.

Lyne, V.D., Hollick, M.: Stochastic Time Variable Rainfall-Runoff Modeling: Proceedings of Hydrology and Water Resources Symposium. Perth, Australia: National Committee on Hydrology and Water Resources of the Institution of Engineers,1979.

(3) We have revised the expression in Ln 249.

During calibration, we used $R^2$ & correlation coefficient of residual sequence (*SCR*) to eliminate the uncertainties caused by the differences in sampling & testing methods of water quality (Yang et al.,2014).

(18) Ln283-299 In the preceding section, the authors mentioned that CN2 has no role. But it appears that runoff curve data was used for calibration. Please clarify.

**Answer:** Thanks for your very thoughtful suggestion.

In the paper, we have added the related contents as follows,

CN2 is a comprehensive parameter that reflects the characteristics of watershed before rainfall. It is mainly affected by the hydrology & soil types, land use types, pre-soil moisture and tillage management measures. CN2 directly affects the surface runoff. The larger the CN2 value, the larger the runoff yield. With the same type of land use, the greater the permeability, the smaller the CN2 value. With the same type of land use, the lower vegetation coverage & rainfall interception ability, the greater the CN2 value. Different regions have different CN2 values, the moist area is the highest, the range of 60~96, while other regions vary greatly. With the same soil types, CN2 value of cultivated land was the highest, followed by grassland and woodland was the lowest.

(19)Results and Discussions:

Similar observations could be made to TP and TN results and discussions. What are the common fertilizers used in the farmlands? Are there pastures and cattle feeding lots? What are the initial soil nutrients content? Did the authors use this information for model parameterization? It is unclear from the methods and discussion.

**Answer:** Thanks for your very thoughtful suggestion.

In the paper, we have added the related contents as follows,

(1) Considering land pattern, rainfall and source of pollutants, the HTRW faces a high risk of pollution from agriculture. Heavy use of fertilizers and soil erosion in the upper of HTRW has led to serious NPS pollution in HTRW.

Fertilization in the HTRW is predominantly with nitrogen, followed by phosphorous and potassium. The heavy use of chemical fertilizers was mainly urea, DAP (diammonium phosphate) & a small amount of N-P-K (Nitrogen-Phosphorus-Potassium mixed fertilizer). Atrazine & Acetochlor were mainly used on dry farmland, and Butachlor was mainly used in paddy fields. Based on the statistical data for 2006-2012, the quantity of fertilizers and Pesticides (such as Methamidophos & Plifenate) applied in the watershed fluctuated annually.

These contents could be found in the Ln 121-129 in the revised version (clean).

(2) Brown soil is widely distributed in the HTRW. We supplied the characteristics of N & P loss under different land use types and fertilization, as shown in Table 2. The thickness of brown soil was 30-50cm in HTRW. The content of organic, TN & TP decreased significantly with the increment of soil depth. Nutrients were mainly found in soils of 0-30cm depth, where TN & TP reserves reached more than 50% of the total reserves in the soil.

**Table 2.** The loss characteristics of N & P under different land use types & fertilization

| Land use type | Soil thickness (cm) | Organic matter content (g/kg) | Unit weight of soil (g/cm$^3$) | Soil particle composition | | | TN (g/kg) | TP (g/kg) |
| --- | --- | --- | --- | --- | --- | --- | --- | --- |
| | | | | Cosmid $\varnothing \leq 0.002$ | Powder $0.002 < \varnothing \leq 0.005$ | Sand $0.005 < \varnothing \leq 2$ | | |
| cultivated field | 0-5 | 24.58 | 1.42 | 21.05 | 57.35 | 21.6 | 0.96 | 0.47 |
| | 5-30 | 18.45 | 1.48 | 24.71 | 24.71 | 18.84 | 0.88 | 0.38 |
| Grassland | 0-5 | 27.6 | 1.18 | 15.97 | 15.97 | 14.58 | 1.25 | 0.58 |

| 5-30 | 21.75 | 1.25 | 20.36 | 20.36 | 21.5 | 1.02 | 0.42 |

Reference:Hao, L.P.: Characteristics of nitrogen and phosphorus losses of rainfall runoff in Liaoning Hunhe Basin, Shenyang Agricultural University, 2012.

The large-scale use of fertilizers (DAP, N:46.4% & N-P-K, N:15%; P2O5:15%; K2O:15%) & livestock and poultry excrement (N:0.5-0.6%; P:0.45-0.6%; K:0.35-0.5%) were the important sources of agricultural NPS pollution. In HTRW, the numbers of pastures and cattle was little, and the excretions of cattle were collected and processed by the farmer.

The excessive or unreasonable application of fertilizers, and the fertilizer utilization rate was not high (the utilization rate of nitrogen is 30% to 60%, and phosphorus is 2% to 25%), resulting in a large number of fertilizer loss. The nutrient content (mainly from agricultural production activities) of soil in HTRW was 1.21t/ha.

The information of initial soil nutrients content & fertilizers was used for model parameterization.

*We tried our best to improve the manuscript and made some changes in the manuscript. These changes will not influence the content and framework of the paper. And here we did not list the changes but marked in red in revised paper (Revision, changes marked).*

*We appreciate for Editors/Reviewers' warm work earnestly, and hope that the correction will meet with approval.*

*Once again, thank you very much for your comments and suggestions.*

---

## Referee Comment (RC1) · Anonymous Referee #1 · 29 Nov 2018

This is an interesting papaer and it has a good scientific soundness. I have some minor comments: Line 10-11 first should come full definition and then abbreviation. Line 34-35 – keywords should not duplicate what is already in the title: e.g. :Agricultural Non-Point Source pollutant loading, Huntai River Watershed. Figure 2 - it is difficult to read text in figures C and D and even more difficult to separate different land use types. I recommend to create them in a same size as B. If not enough space, then I would recommend enlarge them and add to the Supplementary. Secondly, I think that figure 2 title: "The figure was supplied by www.geodata.cn, ...." is not relevant for the figure. I would recommend citing only www.geodata,cn or if needed be then adding some additional information to acknowledgments section. Line 146-47 – Buffer zones

were defined as 1 km in both banks. But did you defined the width of the Buffers? Line 257 – ammonia is not both NH3 and NH4. Line 300 – check fig. 4 title, it seems to me that it is connected with main text. TP loading was reduced only by 10%. Many studies have shown that buffer strips are one of the most efficient measures to reduce P runoff from agriculture. Why is it so low?

———————————————

---

## Author Comment (AC3) · 4 Dec 2018

Anonymous Referee Comments-1 reply

[Figure]

**Reduction Evaluation and Management of Agricultural Non-Point**

**Source Pollutant Loading in the Huntai River Watershed in**

**Northeast China**

Yicheng Fu*, Wenqi Peng, Jinyong Zhao, Xiaoyu Cui

*State Key Laboratory of Simulation and Regulation of River Basin Water Cycle, China Institute*

*of Water Resources and Hydropower Research*

* Corresponding author, E-mail: swfyc@126.com

**Abstract:**

With the raise of environmental protection awareness, applying models to control non- point source (NPS) pollution has become a key approach for environmental protection and pollution prevention and control in China. In this study, we implanted the semi-conceptual model SWAT (Soil and Water Assessment Tool) using information on rainfall runoff, land use, soil and slope. The model was used to quantify the spatial loading of NPS nutrient total nitrogen (TN) and total phosphorus (TP) to the Huntai River Watershed (HTRW) under two scenarios:

without and with projected buffer zones of approximately 1 km within both banks of the Huntai,

Taizi and Daliao river trunk streams and 5 km around the reservoirs. Current land-use types within the buffer zone were varied to indicate the natural ecology and environment. The Nash-

Sutcliffe efficiency coefficient ($E_{NS}$) and $R^2$ for flow and predicted nutrient concentrations (TN

and TP) in a typical hydrological station were both greater than 0.6, and the relative deviation ($|Dv|$) was less than 20%. Under the status quo scenario (SQS), the simulated soil erosion in the

HTRW per year was 811 kg/ha, and the output loadings of TN and TP were 19 and 7 kg/ha, respectively. The maximum loadings for TN and TP were 365 and 260 kg/ha, respectively.

**Fig. 1.**

**Dear Editor,**

We are so appreciated for your letter on our manuscript "Reduction Evaluation and Management of Agricultural Non-Point Source Pollutant Loading in the Huntai River Watershed in Northeast China", Reference No: hess-2018-339. We are also extremely grateful to the comments of anonymous Referee #1 on our manuscript and carefully considered every comment and made cautious revision accordingly. Based on their suggestions, we have answered the questions in detail one by one. If you have any other questions about this paper, I would quite appreciate it if you could let me know them in the earliest possible time.

Most sincerely,

Yicheng Fu, Wenqi Peng, Jinyong Zhao, Xiaoyu Cui
First Contact: Yicheng Fu, swfyc@126.com

Corresponding author:
Name: yi-cheng FU
E-mail: swfyc@126.com
$30^{th}$, Nov. 2018

**Fig. 2.**

---

## Referee Comment (RC2) · Anonymous Referee #2 · 10 Jan 2019

First I don't think the manuscript is novel enough for this journal. Second, the buffers simulated are not realistic so I don't see the benefit of simulating scenarios that are not possible. Third, the model has some fundamental flaws. See below for more details. I didn't review much past the model set up and results. If these are set up incorrectly the results are not worthwhile.

Line 13: change rainfall runoff to precipitation Line 62: specify version. The number of equations changes from version to version. Line 71: GDP is not relevant Line 72: What do you mean by urbanization rate? Looking at Figure 2, it looks like there are a few reservoirs within the watershed. You can't model a watershed with dams unless

you use the reservoir outflow as an input into SWAT. How was this handled? Remove the "The" from each item in the legend. The land use and soil legends are too small. Lines 146-147: Does a 1 km and 5 km buffer sound reasonable and realistic? To me it does not. Line 158: For such a large waters, the number of subbasins and HRUs is quite small especially for the large number of landuses and soil types. Line 178: This is incorrect. It is based on land use, soil type and slope. Line 182: You state that with a threshold of 0, there are 184 HRUs just like you stated in line 158. Then you go on to state that you used a threshold of 5%, 8% and 15%. This would decrease the number of HRUs. Line 189: What about min and max temp? Line 190: What were the results of the crop irrigation time? This varies greatly across a watershed. Line 243: This is too many land uses. You would have many more HRUs. Line 255: What do you mean you simulated rainfall? In line 253 you stated that you had rainfall data from 76 stations. Line 286: These are both downstream of reservoirs Line 362: How many samples do you have from each site? Did you use Loadest or some other program to estimate the loads for days where you didn't have data? How did you compare simulated to monthly concentrations?

---

## Author Comment (AC4) · 19 Jan 2019

Dear Editor, We are so appreciated for your letter on our manuscript "Reduction Evaluation and Management of Agricultural Non-Point Source Pollutant Loading in the Huntai River Watershed in Northeast China", Reference No: hess-2018-339. We are also extremely grateful to the comments of anonymous Referee #2 on our manuscript and carefully considered every comment and made cautious revision accordingly. Based on their suggestions, we have answered the questions in detail one by one. If you have any other questions about this paper, I would quite appreciate it if you could let me know them in the earliest possible time.

[Figure]

Most sincerely,

Yicheng Fu, Wenqi Peng, Xiaoyu Cui, Jinyong Zhao First Contact: Yicheng Fu, swfyc@126.com

Corresponding author: Name: yi-cheng FU E-mail: swfyc@126.com 17th, Jan. 2019

Please also note the supplement to this comment:
https://www.hydrol-earth-syst-sci-discuss.net/hess-2018-339/hess-2018-339-AC4-supplement.pdf

**Supplement:**

**Dear Editor,**

We are so appreciated for your letter on our manuscript "Reduction Evaluation and Management of Agricultural Non-Point Source Pollutant Loading in the Huntai River Watershed in Northeast China", Reference No: hess-2018-339. We are also extremely grateful to the comments of anonymous Referee #2 on our manuscript and carefully considered every comment and made cautious revision accordingly. Based on their suggestions, we have answered the questions in detail one by one. If you have any other questions about this paper, I would quite appreciate it if you could let me know them in the earliest possible time.

Most sincerely,

Yicheng Fu, Wenqi Peng, Xiaoyu Cui, Jinyong Zhao
First Contact: Yicheng Fu, swfyc@126.com

Corresponding author:

Name: yi-cheng FU

E-mail: swfyc@126.com

17th, Jan. 2019

**Additive list**

We have studied the valuable comments from you, the assistant editor and reviewers carefully, and tried our best to revise the manuscript. The point to point responds to the reviewer's comments are listed as following.

**Reviewer's Responses to Questions**

(1) First I don't think the manuscript is novel enough for this journal. Second, the buffers simulated are not realistic so I don't see the benefit of simulating scenarios that are not possible. Third, the model has some fundamental flaws. See below for more details. I didn't review much past the model set up and results. If these are set up incorrectly the results are not worthwhile.

**Answer:** Thanks for your very thoughtful suggestion.

The NPS pollution is prone to cause in dry farmland, paddy, rural & urban areas. The SWAT model has been applied to study NPS in China by numerous research literature, they were mainly focuses on scenario simulation of NPS pollution and management in agricultural areas with rich hydrological and meteorological data. The basic monitoring data of HTRW were deficient; we selected the SWAT as the feasible method to access NPS pollutant loading in watershed level. We applied certain practices based on EPS to reduce the NPS pollutant loading in the Hunhe River, Taizi River and Daliao River watershed. The status quo scenario and EPS were used to calculate the output of NPS pollutant production. The output of NPS pollutant production, the loading intensities of TN & TP was reduced by 21.9%, 25.9% and 10.4% compared with the status quo scenario, respectively. In different regions of NPS pollutant loading in the HTRW changes greatly, and the pollutant loading intensity of different nutrients in the same region is slightly different. Land eco-restoration and land development mode adjustment measures should be practiced reducing NPS pollutant loading of cultivated land.

In order to increase the readability of the paper, we reduced the number of pictures, and increased the number of tables to describe the reduction of agricultural NPS pollution loading. The spatial distribution of the mean annual TP and TN loading in the HTRW were 19, and 7 kg/ha, respectively. The region with a high NPS pollution loading is located in the middle and lower the HTRW, which included the urbanization and population density highly areas of Shenyang, Liaoyang and Anshan. Under the EPS, the TN and TP per unit area were 14, and 6 kg/ha, respectively. The output of NPS pollutant production, the loading intensities of TN & TP was reduced

by 21.9%, 25.9% and 10.4% compared with the status quo scenario, respectively. The NPS pollution occurring within different sub-basins and regions located in the watersheds varied greatly, and the loading intensities of different pollutant types in the given sub-basin were slightly different. Land eco-restoration measures should be implemented to control agricultural NPS pollution from croplands. Therefore, SWAT simulation results provide a reference for the prevention of agricultural NPS pollution in agricultural watersheds.

At present, the Liaoning Liaohekou National Nature Reserve (http://lnsthkgjjzrbhqglj.shidi.org/coohome/coserver.aspx?uid=lnsthkgjjzrbhqglj&sid=20393&clid=5B70C87692924C399BD5A1504571F993&t=66, 121°28′24.58″---121°58′27.49″E, 40°45′00″--41°05′54.13″N) has been completed. The HTRW (40°27'~42°19'N, 121°57'~125°20'E) is situated in the Liaoning province (Northeast China), and the river basin area is $2.73\times10^4$ km$^2$, which comprises approximately 1/5 of the Liaoning province (Fig.2). The establishment of protected areas effectively reduces pollutants. The protected area takes full advantage of the buffer zone. Therefore, the buffers simulated are realistic in the HTRW, and the benefit of simulating scenarios is possible.

(2) Line 13: change rainfall runoff to precipitation.

**Answer:** Thanks for your very thoughtful suggestion.

We have revised the rainfall runoff to precipitation runoff.

(2) Line 62: specify version. The number of equations changes from version to version.

**Answer:** Thanks for your very thoughtful suggestion.

We have changed the contents as follow

The SWAT model's main body contains 80 mathematical equations and 530 intermediate variables.

(3) Line 71: GDP is not relevant.

**Answer:** Thanks for your very thoughtful suggestion.

We have deleted the sentence.

(4) Line 72: What do you mean by urbanization rate? Looking at Figure 2, it looks like there are a few reservoirs within the watershed. You can't model a watershed with dams unless you use the reservoir outflow as an input into SWAT. How was this handled? Remove the "The" from each item in the legend. The land use and soil legends are too small.

**Answer:** Thanks for your very thoughtful suggestion.

We have deleted the sentence of "and the urbanization rate was almost 75%".

By 1989, 689 large, medium and small reservoirs had been built in the Liaohe River Basin, with a total storage capacity of 13.80 billion m$^3$. In the HTRW, there are 4 large reservoirs, which are Dahuofang Reservoir (located in the middle of Hunhe river), Guanyinge Reservoir (located in the upstream of Taizihe river), Shenwo Reservoir (located in the middle of Taizihe river), and Tanghe Reservoir (located in the middle of Tanghe river). Therefore, are many reservoirs in the HTRW. In the SWAT model, we used the reservoir outflow as an input into SWAT.

We have deleted "the" from each item in the legend.

We have enlarged land use and soil legends.

(5) Lines 146-147: Does a 1 km and 5 km buffer sound reasonable and realistic? To me it does not.

**Answer:** Thanks for your very thoughtful suggestion.

The 1 km and 5 km buffer were reasonable and realistic.

(6) Line 158: For such a large waters, the number of sub basins and HRUs is quite small especially for the large number of land uses and soil types.

**Answer:** Thanks for your very thoughtful suggestion.

The downstream of Hunhe River, Taizi River, and Daliao River has little change in terrain, the direction of water flow is single, and the source of contaminant is relatively stable. Therefore, some small calculation units are combined during the calculation process to reduce calculation time and improve operating efficiency.

(7) Line 178: This is incorrect. It is based on land use, soil type and slope.

**Answer:** Thanks for your very thoughtful suggestion.

We have revised the sentence as followed,

Hydrological response unit demarcation is based on land use, soil type and slope.

(8) Line 182: You state that with a threshold of 0, there are 184 HRUs just like you stated in line 158. Then you go on to state that you used a threshold of 5%, 8% and 15%. This would decrease the number of HRUs.

**Answer:** Thanks for your very thoughtful suggestion.

In Line 158 "Hunhe River, Taizi River, and Daliao River sub-catchments were delineated into DEM and river system and further divided by 29 small calculation modules based on 184 HRUs". And in Line 182 "The area threshold percentages for land use, soil and slope were 5%, 8%, and 15%, respectively". There is no correlation between the two.

(9) Line 189: What about min and max temp?

**Answer:** Thanks for your very thoughtful suggestion.

The min and max temp is -30℃and 40℃, respectively.

(10) Line 190: What were the results of the crop irrigation time? This varies greatly across a watershed.

**Answer:** Thanks for your very thoughtful suggestion.

We have added the related content,

such as crop irrigation time (late April and early May) and water.

(11) Line 243: This is too many land uses. You would have many more HRUs.

**Answer:** Thanks for your very thoughtful suggestion.

We delineated land-use types into 27 categories. The main type of HTRW land use and land cover change is forest (including orchard, 49%), dry land (24%), rice paddy (15%), urban land (vacant land, 8%), unused land (uncultivated land, 3%) and grassland (1%). We have combined different land use types into six common types. Therefore, the manuscript is mainly divided into 184 calculation units for the calculation of pollutants for the six land types. The number of calculation units is reasonable.

(12) Line 255: What do you mean you simulated rainfall? In line 253 you stated that you had rainfall data from 76 stations.

**Answer:** Thanks for your very thoughtful suggestion.

To reduce ambiguity, we deleted the expression of "We used meteorological monitoring data to simulate rainfall and evaporation".

(13) Line 286: These are both downstream of reservoirs.

**Answer:** Thanks for your very thoughtful suggestion.

The runoff data series of these two hydrological stations are relatively complete, and the downstream runoff changes can reflect the overall runoff variation of the basin. These two hydrological stations are also the key monitoring sections of the basin, which can reflect the overall spatial and temporal changes in the water volume of the basin.

(14) Line 362: How many samples do you have from each site? Did you use Loadest or some other program to estimate the loads for days where you didn't have data? How did you compare simulated to monthly concentrations?

**Answer:** Thanks for your very thoughtful suggestion.

There are 6 samples of each site. Samples were obtained during the wet season, the wet season, intermediate season, and dry season. Two samples are set for each water period.

We used Loadest to estimate the loads for days where we didn't have data.

The Xingjiawopeng, Xiaolinzi and Tangmazhai Hydrological stations had only the TN data during the study period; therefore, Beikouqian was selected for the validation curves, and the TN $E_{NS}$ and $R^2$ were 0.64 and 0.78, and the TP $E_{NS}$ and $R^2$ were 0.60 and 0.75, respectively (Figs. 6 a and b). The $E_{NS}$ and $R^2$ for the Xingjiawopeng, Xiaolinzi and Tangmazhai hydrological stations were 0.62 and 0.73, 0.61 and 0.72, and 0.62 and 0.77, respectively. The values of all $R^2$ were higher than 0.7, confirming that the SWAT could be used for water quality simulation in HTRW. The simulated TN and TP have a certain synchronization with the measured changes of TN and TP in each month. The variation law of simulated N and P content is not much different from the measured value, and the model has good workability.

*We tried our best to improve the manuscript and made some changes in the manuscript. These changes will not influence the content and framework of the paper. And here we did not list the changes but marked in red in revised paper (Revision, changes marked).*

*We appreciate for Editors/Reviewers' warm work earnestly, and hope that the correction will meet with approval.*

*Once again, thank you very much for your comments and suggestions.*

---

## Author Comment (AC5) · 19 Jan 2019

[revised manuscript text omitted]

---

## Author Comment (AC6) · 19 Jan 2019

[revised manuscript text omitted]